## Comment

palaeontology/evolution

dinosaurs, phylogenetic comparative methods, generalized linear mixed model, diversification rate, diversification decline

**Author for correspondence:**
Manabu Sakamoto
e-mail: msakamoto@lincoln.ac.uk

# Strong support for a heterogeneous speciation decline model in Dinosauria: a response to claims made by Bonsor et al. (2020)

Manabu Sakamoto[1], Michael J. Benton[2] and Chris Venditti[3]

[1]School of Life Sciences, University of Lincoln, Lincoln, UK
[2]School of Earth Sciences, University of Bristol, Bristol, UK
[3]School of Biological Sciences, University of Reading, Reading, UK

 MS, 0000-0001-6447-406X; MJB, 0000-0002-4323-1824;
CV, 0000-0002-6776-2355

Through phylogenetic modelling, we previously presented strong support for diversification decline in the three major subclades of dinosaurs (Sakamoto et al. 2016 Proc. Natl Acad. Sci. USA **113**, 5036–5040. (doi:10.1073/pnas.1521478113)). Recently, our support for this model has been criticized (Bonsor et al. 2020 R. Soc. Open Sci. **7**, 201195. (doi:10.1098/rsos.201195)). Here, we highlight that these criticisms seem to largely stem from a misunderstanding of our study: contrary to Bonsor et al.'s claims, our model accounts for heterogeneity in diversification dynamics, was selected based on deviance information criterion (DIC) scores (not parameter significance), and intercepts were estimated to account for uncertainties in the root age of the phylogenetic tree. We also demonstrate that their new analyses are not comparable to our models: they fit simple, Dinosauria-wide models as a direct comparison to our group-wise models, and their additional trees are subclades that are limited in taxonomic coverage and temporal span, i.e. severely affected by incomplete sampling. We further present results of new analyses on larger, better-sampled trees ($N = 961$) of dinosaurs, showing support for the time-quadratic model. Disagreements in how we interpret modelled diversification dynamics are to be expected, but criticisms should be based on sound logic and understanding of the model under discussion.

# 1. Introduction

Recently, Bonsor et al. [1] criticized our selection of a model of diversification in dinosaurs that supported a long-term decline in the rate at which speciation events accumulated across the dinosaur tree of life through time (time-quadratic model) [2]. Core to their criticism, Bonsor et al. [1] applied our analytical approach to nine additional trees that were published subsequently to our 2016 paper [2], claiming that the time-quadratic model is not as well supported as we originally reported. They go on to list several criticisms of our choices in selecting the time-quadratic model as the preferred model of diversification. Here, we highlight some of what we believe to be misconceptions and misinterpretations that Bonsor et al. have made about our work and we also explain our concerns about their own analyses using the nine additional trees. Additionally, we fitted and compared the three models on a sample of 100 meta-trees of dinosaurs ($N = 961$) [3] and demonstrate that the group-wise time-quadratic model is still strongly supported over the time-square root model and time-linear model.

# 2. Models of speciation in dinosaurs analysed by Sakamoto et al. (2016)

First, we will clarify the models we fitted and compared in our 2016 study [2]. We modelled how speciation events accumulated across phylogeny through time by taking the number of nodes from the root to each tip of the phylogeny ($N_{\mathrm{Nodes}}$) as the response variable in a phylogenetic Poisson model, fitted through a Bayesian generalized linear mixed model (MCMCglmm [4]).

In order to test three alternative hypotheses relating to how nodes accumulate (fig. 1 from [2] but also reproduced as fig. 1 in [1]), we compared three models, each differing in how the temporal effects were modelled as the predictor variables: Model A, the time-linear (null) model, $N_{\mathrm{Nodes}} = \mathrm{Time}$; Model B, the time-square root model, $N_{\mathrm{Nodes}} = \sqrt{\mathrm{Time}}$; and Model C, the time-quadratic model, $N_{\mathrm{Nodes}} = \mathrm{Time} + \mathrm{Time}^2$.

Model A is where $N_{\mathrm{Nodes}}$ steadily accumulates through time at a constant rate, Model B is where $N_{\mathrm{Nodes}}$ accumulates rapidly initially but then slows down towards an asymptote and Model C is where $N_{\mathrm{Nodes}}$ accumulates rapidly initially but slows down and is allowed to decrease further in a downturn.

We fitted this set of three models in two ways: (i) as a single set of parameters across the entirety of Dinosauria; and (ii) as separate sets of parameters for five subclades (Hadrosauriformes, Ceratopsidae, Ornithischia [without hadrosaurs and ceratopsids], Sauropodomorpha and Theropoda) in a single model framework (group-wise model).

Thus, we fitted six models in total: Model A-Dino, Dinosauria time-linear model; Model B-Dino, Dinosauria time-square root model; Model C-Dino, Dinosauria time-quadratic model; Model A-5G, 5-group time-linear model; Model B-5G, 5-group time-square root model; and Model C-5G, 5-group time-quadratic model. In actuality, we fitted and compared various other iterations and grouping structures (such as the 3-group model), but these six are the models relevant to our discussion here and in our original paper [2].

We compared model fit using deviance information criterion (DIC) and selected the model with the lowest DIC score, at least 4 less than the next lowest DIC score, as the preferred model [2]. We found that Model C-5G had the lowest DIC score and we thus selected this model as the preferred model.

Now that we have covered our model fitting and comparisons, we can proceed to address the criticisms made by Bonsor et al. [1].

# 3. A response to the claim that heterogeneous speciation dynamics was not accounted for

Bonsor et al. [1] criticize us for ignoring the presence of heterogeneity in speciation dynamics across several subclades. As evident from our review of models above, this criticism is, in our view, unjustified. Our selected model was Model C-5G, the one where speciation dynamics were specifically allowed to vary across five major dinosaur groups [2]. As the time-quadratic effects were not detected for hadrosauriforms and ceratopsids, we can interpret this to mean that these two subclades were not in 'decline'. Crucially, even after accounting for such heterogeneity, strong time-quadratic effects were detected in the other three clades, which constitute the vast majority of dinosaurian lineages (hadrosauriforms and ceratopsids collectively account for only 14% of the taxa). Thus, the selected

model, on which we based our interpretations, specifically accounts for differences in group-wise speciation dynamics.

This fact was communicated clearly, even stated in the abstract to our paper:

> The only exceptions to this general pattern are the morphologically specialized herbivores, the Hadrosauriformes and Ceratopsidae, which show rapid species proliferations throughout the Late Cretaceous instead. Our results highlight that, *despite some heterogeneity in speciation dynamics*, dinosaurs showed a marked reduction in their ability to replace extinct species with new ones, making them vulnerable to extinction and unable to respond quickly to and recover from the final catastrophic event. [2].

## 4. A response to the claim that models were selected through parameter significance

Bonsor *et al.* criticize us for basing our selection of the time-quadratic model on parameter significance instead of DIC scores. This is inaccurate. While we state in the methods section that

> In the case where multiple models had nonsignificant differences in model fit (i.e. ΔDIC < 4), we inspected the significance of model parameters and selected the model with significant covariates (i.e. nonsignificant covariates were removed) [2],

this applies to the removal of covariates (in a backward elimination approach; note the final statement of the quote above, 'nonsignificant covariates were removed'). 'Covariates' here largely refer to predictor variables aside from the main time-dependent variables. These include sea level, competition, sampling, etc. Additionally, we used parameter significance to determine whether certain grouping parametrization was more informative over another, namely in how the slopes and intercepts were modelled for the hadrosauriforms and ceratopsids (see SI from [2]).

Therefore, this approach does not apply to the selection of the time-quadratic model over the time-square root model. In the 5-group models, Model C-5G (5-group time-quadratic model) is generally selected over Model B-5G (5-group time-square root model) based on DIC comparisons (also see results of a set of new analyses below). Where there is no significant difference between Models B and C, it is only in comparisons between the Dinosauria-wide models (Model B-Dino versus Model C-Dino). The following is from our 2016 paper (emphases added here):

> Although the square root and quadratic models were not significantly different for the *Dinosauria as a whole* (both of which were significantly better than those in the linear model), *the quadratic model was generally better than the square root model in both the three- and five-group models* [except in the tree by Lloyd *et al.*] [2],

but even then, model selection is still based on DIC in the Dinosauria-wide models with respect to the Time-related variables. Thus, *the claim that we selected our model based on parameter significance instead of DIC is not correct*.

Furthermore, because of how the models are formulated, it is not even possible to select the time-quadratic model over the time-square root model based on parameter significance. Recall, Model B is $N_{Nodes} = \sqrt{Time}$ while Model C is $N_{Nodes} = Time + Time^2$. Note that these models are not nested and both sets of predictors are significant, i.e. $\sqrt{Time}$ is significant as is $Time^2$ (see SI from [2]). Hence, it is impossible to select one over the other based on parameter significance. If in the case that $Time^2$ is significant while $\sqrt{Time}$ is not, then that would mean that the quadratic model is indeed the better fit than the square root model.

As a side note, the lack of a significant difference in 'the tree by Lloyd *et al.*' refers to one specific tree for which branch lengths were scaled using the first appearance dates (FADs) for all the tips. This represents an extreme case, which is highly unlikely to be the true distribution of species ages. Since tip ages are associated with date ranges, this case would be one out of an astronomical number of combinations of randomly sampled ages where all sampled ages just happen to be FADs. The Lloyd *et al.* supertree [5] is also the smallest of the trees we analysed ($N = 420$ compared with 614 of the Benson *et al.* trees [6]) and given that incomplete sampling is considered to affect estimating diversification dynamics, it is not surprising that such an extreme branch length scaling would result in such uncertainties.

## 5. Models and trees of Bonsor *et al.* (2020) are not comparable to those of Sakamoto *et al.* (2016)

Bonsor *et al.* fitted and compared the three models (Models A–C) in nine additional trees as well as the three trees we analysed in our 2016 paper [1,2]. Their results show ambiguity in model selection based on

DIC, with the time-quadratic model rarely being selected, even in the reanalyses of our three trees. However, there are largely two problems with their analyses: (i) they equate the Dinosauria-wide model with the 5-group model and refute the latter through reanalyses of the former; and (ii) their nine additional trees are substantially smaller with severely limited taxonomic coverage and are, therefore, not comparable to our trees and subclades therein.

## 5.1. The Dinosauria-wide model is not equivalent to the 5-group model

As stated above, our original analyses did not result in significant differences between Models B and C when fitted as a single set of parameters across the Dinosauria as a whole. However, the time-quadratic model was generally selected over the time-square root model when group-wise effects were modelled [2]. Bonsor *et al.* did not fit and compare the 5-group models, but they proceed to argue that our selection of the 5-group time-quadratic model is wrong, based on their results from the Dinosauria-wide models [1].

However, we did not select Model C-Dino, but rather, Model C-5G, the 5-group time-quadratic model. Thus, demonstrating that the time-quadratic model is not better than a time-square root model in Dinosauria does not negate the fact that the 5-group time-quadratic model is a better fit to the data than the 5-group time-square root model (or any of the Dinosauria-wide models).

It is not evident from the text of their article that Bonsor *et al.* did not fit a 5-group model. This is only revealed once their R [7] script is reviewed. Their R scripts and data are not readily available through the journal's supplementary materials but only through the authors' git-hub repository. Git-hub repositories, while having great utility in sharing data and code, are not ideal for users unfamiliar with the system. Additionally, given that the model formulation can only be found in one of their R functions, a certain level of proficiency in R is expected of the reader just to confirm the model formulation. This information should have been made transparent in their publication.

## 5.2. Subclade trees are not equivalent to whole trees

The taxonomic scope of the nine trees analysed by Bonsor *et al.* are as follows: Arbour dataset, Ankylosauria ($N = 57$) [8]; Thompson dataset, Ankylosauria ($N = 50$) [9]; Raven dataset, Stegosauria ($N = 23$) [10]; Chiba dataset, Ceratopsia ($N = 30$) [11]; Mallon dataset, Chasmosaurinae ($N = 27$) [12]; 'CruzadoC' dataset, Hadrosauriformes ($N = 62$) [13]; Carballido dataset, Sauropodomorpha ($N = 87$) [14]; 'GonzalezR' dataset, Titanosauriformes ($N = 76$) [15]; and Cau dataset, Coelurosauria ($N = 141$) [16] (figure 1).

Despite constituting the bulk of their trees, none of the ornithischian subclades analysed by Bonsor *et al.* can be considered equivalent to the whole clade Ornithischia (all ornithischians excluding hadrosauriforms and ceratopsids) modelled in our analyses [2]. Such sparsely or narrowly sampled subclades cannot be treated as the equivalent of a more widely and more completely sampled tree (figure 1). Rather surprisingly, three of the datasets analysed by Bonsor *et al.* (Chiba, 'CruzadoC', Mallon) are focused on the two subclades (Hadrosauriformes and Ceratopsidae) for which we did not find significant time-quadratic effects. Therefore, a lack of a good fit for a time-quadratic model in these three datasets is expected—that is, three out of six of their ornithischian datasets are by default not appropriate as benchmarks for determining if a time-quadratic effect can be detected. Additionally, the other three ornithischian datasets (Arbour, Thompson, Raven) are ankylosaurs and stegosaurs, which are both subclades of Thyreophora, not remotely representative of Ornithischia as a whole.

Similarly, the dataset representing Theropoda in the analyses of Bonsor *et al.* (Cau dataset) only covers Coelurosauria [16], which while making up a large portion of Theropoda, is a derived subclade, restricted to the latter half of the theropod evolutionary history, starting some time in the Mid- to Late Jurassic period (figure 1). One would not expect to detect the same slowdown or downturn effects within the period of time covered by this tree, when compared with our tree, which covers the entirety of Theropoda and their whole evolutionary history, from the Late Triassic to the Cretaceous. One of the two sauropod trees analysed by Bonsor *et al.* ('GonzalezR' dataset [15]) covers Titanosauriformes, which much like Coelurosauria, is a derived, predominantly Cretaceous subclade within Sauropoda.

The only tree out of the set analysed by Bonsor *et al.* that can be considered a valid comparison was the Carballido dataset [14], which comprises a taxonomic sample ($N = 87$) that spans a similar range to our Sauropodomorpha subclade [2] (figure 1). Unsurprisingly, this tree is the only one analysed by Bonsor *et al.* that strongly supported a time-quadratic model compared with the time-square root model [1].

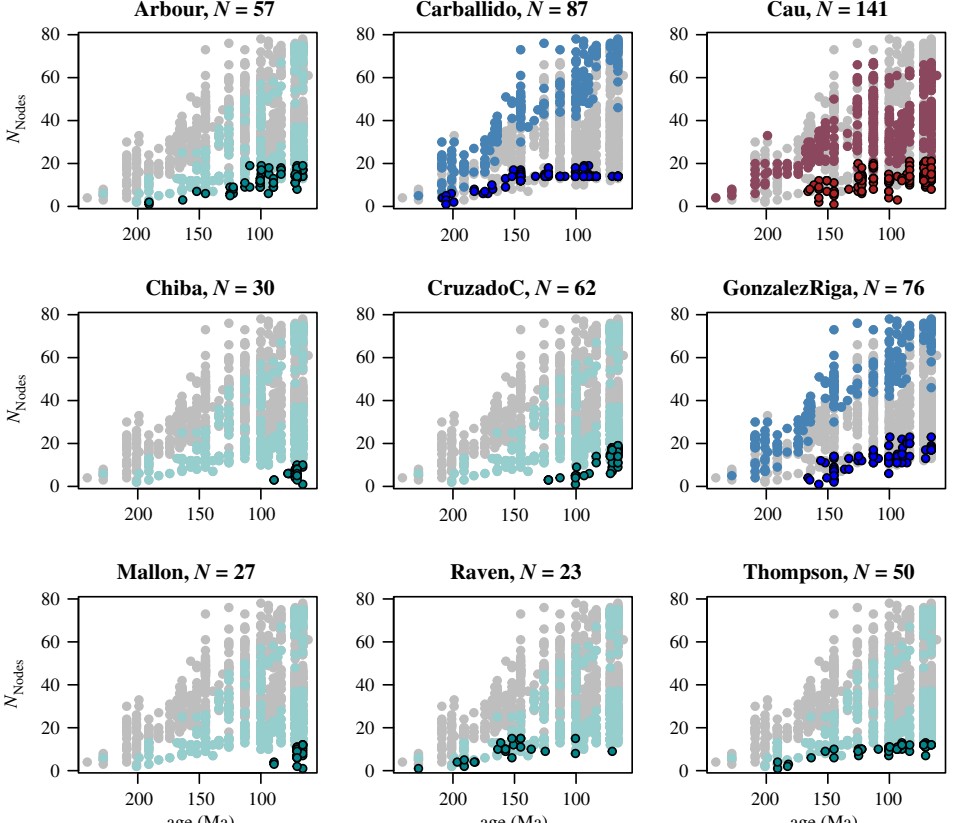

**Figure 1.** $N_{Nodes}$ is plotted against Time (here expressed as the last appearance dates in million years ago) for the meta-tree of Dinosauria ($N = 961$) [3] in grey. $N_{Nodes}$ and Time from each of the nine trees used by Bonsor *et al.* [1] are superimposed in colours corresponding to Ornithischia (green), Sauropodomorpha (blue) and Theropoda (red). Lighter shades are the subclades in the larger tree that each of the nine trees are supposed to represent and are only shown for reference. This is especially important for the hadrosauriform and ceratopsian (Chiba, 'CruzadoC' and Mallon) datasets where despite their narrow taxonomic coverage, Bonsor *et al.* used them to represent the wider ornithischian clade.

This demonstrates that once taxonomic coverage increases to a comparable level, even Bonsor *et al.* recovered results that were qualitatively identical to ours.

Models fitted to small subclades are not directly comparable to those fitted to our larger tree. As our group-wise model was fitted across the entire Dinosauria tree, with the model parameters estimated separately and simultaneously for individual subclades, DIC comparisons and model selection were conducted on the entire Dinosaur tree, not based on individual subclades. Thus, our model cannot be compared with models fitted on individual subclades.

# 6. To estimate the intercept or not

Bonsor *et al.* [1] fitted models where the intercept is fixed at 1 on the basis that speciation events at Time = 0 is technically 1 if counting the root node. There are two issues with this. First, as we discussed in Sakamoto *et al.* [2], fixing the intercept to the theoretical starting value of 1 at Time = 0 (at the root of the tree) makes a strong assumption that the earliest divergence occurred at the inferred root age of the tree. As the root ages of our trees were based on external independent sources based on best estimates from the fossil record, these are subject to error, especially since the earliest fossil record of dinosaurs is still relatively poor. The earliest dinosaurs and dinosauriforms were largely contemporaneous (on a geological time scale) so the origin of Dinosauria cannot be reliably estimated beyond the oldest members of both dinosaurs and dinosauriforms. That is, most fossil estimates for the origin of dinosaurs at around 247–248 Myr will most likely be minimum ages—it is possible that the true divergence date was older. Estimating the intercept on the other hand will add an offset based on the data, which can be interpreted as accounting for uncertainty in the amount of time elapsed since the true origination date for the dinosaurian phylogenetic tree. Therefore, while

theoretically the intercept should be forced through the origin, this only applies to cases where we are confident of the root age, such as trees generated from laboratory-manipulated experiments (on bacteria or viruses) or simulated trees.

Second, if the theoretical number at Time = 0 truly is 1, then the intercept of a Poisson model should be fixed to a value of 0, not 1. This is because the Poisson model uses a log-link so the expected outcome for a time-quadratic model is

$$\log E[N_{\text{Nodes}}|\text{Time}] = \beta_0 + \beta_1 \text{Time} + \beta_2 \text{Time}^2$$

which can be rewritten as

$$E[N_{\text{Nodes}}|\text{Time}] = \exp(\beta_0 + \beta_1 \text{Time} + \beta_2 \text{Time}^2).$$

Since at Time = 0, $E[N_{\text{Nodes}}|\text{Time}] = \exp(\beta_0)$, if we fix the intercept $\beta_0$ to 1, then $E[N_{\text{Nodes}}|\text{Time}] = \exp(1) = 2.72$, so in order for $E[N_{\text{Nodes}}|\text{Time}]$ to be 1, $\beta_0$ should be fixed to 0, i.e. $E[N_{\text{Nodes}}|\text{Time}] = \exp(0) = 1$, which is the model formulation we actually fitted and compared with in our original 2016 analyses [2].

# 7. New analyses of a sample of meta-trees of Dinosauria ($N = 961$) support the speciation decline model

For completeness, we modelled $N_{\text{Nodes}}$ through time on a sample of recent meta-trees of Dinosauria [3], which covers 961 species. We selected 100 topologies at random from the sample of 1000 trees provided in the SI [3]. We scaled the branches to cover both extremes of branch lengths, the shortest possible tree and the longest possible tree. We analysed both sets of trees (i.e. 200 trees in total).

We fitted the three competing models for each tree topology (Models A–C; 3 models × 2 sets of 100 trees = 600 models total). We estimated separate model parameters for each of the five subclades, Ceratopsidae, Hadrosauriformes, Ornithischia (non-ceratopsid, non-hadrosaurid), Sauropodomorpha and Theropoda. For Ceratopsidae and Hadrosauriformes, Time$^2$ and $\sqrt{\text{Time}}$ were not modelled and instead only the Time effects were modelled. To account for the potential effects of incomplete sampling, we included a measure of fossil sampling, log number of fossil occurrences ($N_{\text{Occ}}$). $N_{\text{Occ}}$ were tallied up from fossil occurrences for Dinosauria downloaded from the Paleobiology Database [17] (accessed 5 March 2019 using keywords Dinosauria restricting age to Mesozoic and excluding ichnofossils). We compared model fit using DIC and selected the model with a DIC score that is at least 4 less than the alternative model (i.e. ΔDIC > 4).

In both sets of 100 trees analysed, the difference in DIC between the 5-group time-quadratic and 5-group time-square root models was at least 9 and 10 for the shortest and longest trees, respectively (with the median being 13 and 15, respectively) in favour of the 5-group time-quadratic model. The 5-group time-linear model was firmly rejected in every comparison. Therefore, we provide here additional support for the 5-group time-quadratic model over the 5-group time-square root and the 5-group time-linear models [2].

# 8. Does high diversity to the end of the Cretaceous mean dinosaurs were not in decline?

Bonsor *et al.* draw from the literature to demonstrate that species richness, morphological diversity (commonly referred to as disparity) [18] as well as diversity in ecological niches [19] all indicate that dinosaurs maintained high diversity to the end of the Cretaceous period. While we do not dispute such observations, we simply highlight that, even if standing diversity appears to be maintained at a constant level, that does not mean that the net-speciation rate is zero. Diversification rates implied by differences in time-binned average values are susceptible to the undesired effects of phylogenetic non-independence. That is, changes in diversity (whether taxonomic diversity, morphological diversity or ecological diversity) from one time interval to another cannot be taken at face value. Consider a case where a high-diversity clade (e.g. Rodentia at over 2000 species) and a low-diversity clade (e.g. Perissodactyla at around 17 species) coexist across multiple time intervals. A diversity increase of 10 species over 1 Ma may not be high for the former clade but will be exceptional for the latter. In fact, a mere 10-species increase over 1 Ma may represent a decrease in speciation rates for the former clade given the probability of speciation

occurring just by chance owing to the sheer number of living species. That is, the underlying expected probability of speciation must be considered when inferring speciation rates.

## 9. Does incomplete sampling affect diversification models?

While Bonsor *et al.* rightly refer to the incompleteness of the fossil record as a potential source of uncertainties in models of diversification, they do not acknowledge that our model accounts for incomplete sampling through the inclusion of relevant covariates [2]. In fact, we tested multiple measures of sampling. We also included a measure of sampling (number of occurrences) in our current seven reanalyses of our models on the sample of 100 $N = 961$ meta-trees [3]. Measures of sampling are often not significant, but even when they are, their inclusion in the model does not affect the time-quadratic effects.

One key feature of our Poisson model of $N_{\text{Nodes}}$ through time is that we are not directly parametrizing diversification rate through a birth–death process—which often incorporates sampling as part of the model parametrization. Instead, our model is a simple relationship between the accumulation of nodes and the passage of time. Since the accumulation of nodes is the response variable, any extrinsic variable that can be hypothesized as having an effect can be modelled as a fixed effect or covariate. This means that if a relationship between $N_{\text{Nodes}}$ and a measure of sampling can be suggested in a regression framework, we would be able to incorporate this into our model. However, Bonsor *et al.* do not do this and we worry it may lead a reader to infer that our model suffers from unacknowledged incomplete sampling, a point (to reiterate) that we addressed in our earlier paper.

## 10. The importance of phylogenetic comparative methods in macro-evolutionary research

Finally, Bonsor *et al.* extend their criticisms to question the utility of phylogenetic methods to 'solve all of the problems that remain' [1]. Phylogenetic approaches are powerful (if not necessary) statistical tools to account for undesired effects of phylogenetic non-independence, but the purpose of their use in macro-evolutionary studies is not to 'solve all of the problems that remain'. Since speciation (e.g. cladogenesis) is inherently a phylogenetic process, any attempts to analyse speciation or diversification dynamics must be conducted in a phylogenetic framework (this also includes attempts to measure diversification rates from diversity across time intervals). Thus, the use of phylogenetic approaches is not a means to solve all remaining problems but is the foundation upon which future development to solve such outstanding problems must be built.

## 11. Conclusion

Here we refute the criticisms made by Bonsor *et al.* [1] of our earlier analysis [2] based on the fact that we consider their criticisms largely to stem from misunderstandings or misinterpretations of our study. Our selected model accounts for heterogeneity, are selected based on DIC scores (not parameter significance), and intercepts were estimated to account for uncertainties in root age. Additionally, their new analyses are not comparable to our models. They fit simple, Dinosauria-wide models as a direct comparison with our group-wise models, and their additional trees are subclades that are limited in taxonomic coverage and temporal span. Thus, arguments based on such analyses are not appropriate. We also demonstrate using 100 large trees ($N = 961$) of dinosaurs that the time-quadratic model [2] is supported when newer and better-sampled trees are used. In closing, we would encourage our colleagues—in all fields—to ensure that, when critiquing others' work, this is done with the fullest understanding and most logical argument possible. We are grateful to have this opportunity to engage with Bonsor *et al.* and encourage all readers to assess the works discussed with this in mind.

Data accessibility. The data and codes for our reanalyses are available through the Open Science Framework at: https://osf.io/uct2p/.

Competing interests. We declare we have no competing interests.

Funding. This work was supported by NERC grant no. NE/P013724/1 to M.J.B.

Acknowledgements. We thank the two anonymous reviewers for their comments that ultimately strengthened this manuscript. We also thank the editors, Anita Kristiansen and Andrew Dunn for their hard work in the peer review process during these difficult times.

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
