## [Peer Review File · Royal Society Open Science]

Review History

RSOS-202143.R0 (Original submission)

Review form: Reviewer 1

Is the manuscript scientifically sound in its present form?

No

Are the interpretations and conclusions justified by the results?

No

Is the language acceptable?

No

Do you have any ethical concerns with this paper?

No

Have you any concerns about statistical analyses in this paper?

Yes

Recommendation?

Major revision is needed (please make suggestions in comments)

Comments to the Author(s)

I have read the comment article "Overwhelming Support for Diversification Decline in Dinosaurs" by Sakamoto et al. The article is a response to a previous paper by Bonsor et al, published last year. I also re-read that paper while conducting this review.

The comment by Sakamoto et al focuses on analysing data across the whole tree of dinosaurs, which includes hundreds of species. In contrast, Bonsor et al focussed on subtrees, and did not include variables to distinguish between the dynamics of diversification in different groups in their models. Sakamoto et al focus on those differences in approach to analysis. Here, Sakamoto et al argue that, once these things are done (large trees, clade-specific model terms), their original hypotheses findings are robustly supported.

They do not mention that Bonsor et al also levelled other criticisms of their original analysis. For example, their choice not to fix the intercept term to theoretically meaningful values. And the argument that the 'diversification decline', and arguments such as 'Late Cretaceous dinosaurs were incapable of speciation' (paraphrasing) do not take into account that some dinosaur subgroups were diversifying very rapidly in the Late Cretaceous, contrary to various statements in the original article by Sakamoto et al (e.g. dinosaurs lost their ability to speciate; paraphrasing). These other aspects seem to be integral to the criticisms of Bonsor et al but are not addressed in the response by Sakamoto et al. Importantly, the decision to not even acknowledge these other criticisms is being potentially misleading to readers of the Sakamoto et al response who may regard their summary as being a complete one. Acknowledging the multiple points of the arguments of Bonsor et al should therefore be a prerequisite of publication.

Sakamoto et al make two main points. The substance of one of these is that Bonsor et al did not use sufficient sample sizes, and did not allow clade-specific dynamics. However for some clades, the analyses of Bonsor et al include potentially similar sample sizes to those of subclades that were allowed to exhibit different diversification dynamics in the analyses of Sakamoto et al. I have provided some more information about this below. The overarching conclusion I have, based on the manuscript as currently written, is that the authors have presented an incomplete account of the nature of sample size, and have not provided sufficient information on effective sample sizes for a reader to really be able to evaluate their arguments. This needs to be addressed so that an informed reader would be able to make a fair comparison. Note, I am not saying that the authors are wrong. Indeed, I'm leaning towards the conclusion that they are right, at least so far as this specific set of model comparisons is concerned. However, there is important information missing that would allow me to fully judge the strength of their arguments.

Title: The title is over-blown and not strictly informative. Whether we consider a decline in diversification rates to be 'overwhelmingly supported' depends on factors other than those covered in the paper. For example, is the approach to testing this hypothesis appropriate in the first place? Is the assumption that nodes in a paleontological phylogeny provide unbiased information about counts of speciation events really justified (note: widespread methods such as the fossilize-birth death model assume this is -not- the case)? There has received essentially no scrutiny of method performance, so we don't really know the answer to these questions, and it's not really appropriate to get into the weeds on that topic in the present context. However, I'd like to see a more circumspect and less bombastic approach to writing, with more the characteristics of reflective science that acknowledges uncertainties. That is important here in the title and elsewhere in the manuscript.

In summary. The authors have a right to explain their position. But that explanation should acknowledge the multiple facets of the work by Bonsor et al rather than picking and choosing. Furthermore, the current manuscript should focus more on communicating information that is necessary for a reader to evaluate the strengths of the argument. And avoid bombastic language. Myself, I am part-way there. But I will be convinced by this once: (1) The authors provide clearer explanations and comparisons of effective sample size. (2) Ideally include a more well-reasoned explanation of their $N = 50$ sample size threshold. And (3) Acknowledge the other points made by Bonsor et al regarding (i) Model interpretation (that some groups show a contrarian pattern) and (ii) Theoretically non-sensible intercept terms. I view these aspects as being a relatively substantial re-tooling of the manuscript.

Detailed comments

>"Recently, Bonsor et al. [1] claimed to have put into question our preferred model of dinosaur diversification (number of speciation events through time) presented by Sakamoto et al. [2], by analysing an additional nine phylogenetic trees that were published subsequent to our study."

>>>This is not the entire substance of the arguments of Bonsor et al. Some aspects of the arguments are not even mentioned in the current article. The decision just to focus on one aspect allows the authors to potentially win that part of the argument. But it leaves other, potentially-important, aspects completely unaddressed. That's OK if they authors do not aim to convince people that they are correct. However, it is also potentially misleading because some readers may regarding the summary provided here as being a complete account of the arguments of Bonsor et al.

>"which are not apparent from their article but evident from their data and codes: they used an overly simplified model formulation that is not comparable to our model"

>>>Ok, agreed.

>"and their analyses are generally based on small sample sizes."

>>>I agree with this in part. However, while reading the current ms I had various questions about the effective sample size of the analyses once terms are included to model different dynamics among subclades. The substance of my main question is as follows: Say I have a tree of 700 species, of which 40 are hadrosauroids. When I include a term to allow different dynamics of diversification in hadrosauroids, what is my effective sample size for inferring diversification dynamics in hadrosauroids? I believe it likely is $N = 40$. However it depends somewhat on model formulation because the model could treat their dynamics as inheriting some aspects from their ancestry, or it could allow them to be completely decoupled. If complete decoupled, then the effective sample size for a subclade is likely to be the number of species sampled in that subclade (actually, a smaller number given the fact of hierarchical autocorrelation in phylogenetic comparative methods). Therefore, we need more information and the comparison needs to be more careful and overall transparent.

>>>The overarching conclusion I have, based on the manuscript as currently written, is that the authors have presented an incomplete account of the nature of sample size, and have not provided sufficient information on effective sample sizes for a reader to really be able to evaluate there arguments.

>"Sakamoto et al modelled separate effects of Time and Time2 on the individual subclades, Theropoda, Ornithischia, Sauropodomorpha, Ceratopsidae and Hadrosauriformes [2] within a single model framework. "

>>>Yeah, so basically the question here is whether I should expect different answers for sauropodomorph dynamics if I fit the model to just sauropodomorphs on their own? If the dynamics of sauropodomorph diversification in the 'whole Dinosauria' model are genuinely decoupled from those of other groups then the appropriate comparison of sample sizes would be between the count of sauropodomorphs included in each analysis. -Not-, as implied here, a comparison of the sample size of all Dinosauria in Sakamoto's original paper, to that for Sauropodomorpha in one of Bonsor et al's analyses. At least, that's where I stand on this topic at the moment. And if that is not the case, then it is the author's job to convince readers. This will require a much more reflective and circumspect approach than is currently manifested in the paper.

>"which, for some reason, is the model formulation that Bonsor et al. [1] chose to test instead"
>"...as Bonsor et al rightly point out (following our explicit and repeated statements...)"

>>>This language comes across as being pejorative and doesn't seem appropriate to me if the job is to convince using reasoned and compelling arguments.

>"When the group-wise model formulation is used... then the differences in DIC are indisputably in favour of the Time2 model, thus providing overwhelming support for this model of diversification through time"

>>>I see what the authors are trying to say here. However, the group-wise model also posits that some of the youngest groups (ceratopsian, hadrosauroids) show a different pattern that goes in the opposite direction. This is relevant to interpretation of this statistical model result. Namely, that the model might then -not- be interpreted as providing 'overwhelming' support for a diversification decline across all Dinosauria. This argument was made by Bonsor et al, and I feel that the authors here have failed to get to grips with it in an appropriately circumspect manner. See e.g. their title and various other overly-strong statements of unilateral interpretation.

>"The additional trees used in their analyses are very small (N = 23-141), with fewer species sampled per clade, compared to those we used (N = 420 and 614).

>>>The text starts 'species per clade', but the N values given for the original study of Sakamoto et al are not 'per clade values'. They are 'all Dinosauria' values. So, of course the numbers are large compared to the numbers used in the analyses of dinosaurs subclasses presented by Bonsor et al. However clearly this is a misleading as currently phrased. See my comment above about explanations of samples sizes and effective samples sizes. I feel it is likely, even after a 'fair' comparison, that the analyses of Sakamoto et al have larger effective sample sizes than the analyses of Bonsor et al. Nevertheless, Sakamoto et al need to demonstrate this in a more transparent way.

>"In fact there is an even large meta tree..."

>>>That's true, it is large. Have the authors looked at the timescale phylogeny and which taxa are controlling the divergence times though? There are various species in there that have essentially unknown phylogenetic affinities (based on comparative anatomical and primary phylogenetic literature). Those taxa sometimes are placed in phylogenetic positions that are deeply nested compared to what you'd guess based on their times of appearance. This has a big effect on the divergence times found by Lloyd et al. Sure, OK, it's a tree that exists and could be analysed. But the fact that many of the divergence times are pushed to surprisingly old ages is probably artifactual and I believe it would tend to bias in analysis in favour of the hypothesis of Sakamoto et al.

>"Larger sample sizes offer more statistical power in estimating parameters in complex models of diversification. Conversely, smaller sample sizes lack the appropriate statistical power to differentiate between competing models. This is especially so for trees with $N < 50$. Simulations have demonstrated that simple evolutionary parameters like heritability"

>>>Sure, I agree with the principle that large sample sizes give more statistical power. However, the text is very generic. It asserts that variation in sample size within the range of the analyses of Bonsor et al will be important in the specific example. But doesn't provide evidence from simulations that are relevant to the current estimation task. Instead, it refers to estimating heritability, which is really quite a different inference problem. It also lacks nuance. For example, greater statistical power is particularly important when effect sizes are small. So if there is 'overwhelming' support (i.e. large effect sizes), then we may not expect this to be as much of a problem. Note that I'm not saying that the authors are wrong. Instead, I'm saying that the number 'N=50' seems to have been chosen arbitrarily without much rationale behind it. I don't think this way of making the argument sets a good example for others to follow.

>>>Also, what is the effective sample size for this type of analysis. Cecile Ané did some nice work showing the effective sample sizes for phylogenetic comparative methods could be much lower than the number of species samples, and that the effective sample size increases initially rapidly when few species have been sampled, and then slows down, with new additions of species adding relatively little new information. Of course, she was looking at a different estimation problem. I only write this here to indicate that it isn't possible to say a priori what sample size is needed without further evaluation of method performance. That evaluation has not been done, to my knowledge, but could provide some of the best support for the authors' preferred hypothesis.

>"In fact, Bonsor and colleagues [1] could not differentiate between the null model and the two competing models in the majority of the trees they analysed, indicating that their sample sizes do not offer sufficient statistical power".

>>>There's no way to approach this other than to say that this reasoning is clearly flawed from the most basic level of statistics. The fact that the null hypothesis cannot be rejected could either reflect low statistical power, as argued here, or -that there is no evidence for the alternative hypothesis. Please remove or re-phrase this specific sentence.

>"Additionally, counting the number of nodes on smaller trees inevitably leads to a narrower range of speciation events (Fig. 1)"

>>>The fact that smaller trees have fewer nodes is self-evident. Please replace 'narrower range' with 'smaller count' or similar.

>"...at least three of their trees suffer from a lack of variation on the main predictor variable, Time"

>>>Agreed.

>"Thus, arguing that one model is not discernible from another, or indeed even from conclusive. In fact, such results suffer from severe sampling biases".

>>>'Bias' refers to systematic effects that would incorrectly prefer one model over others. That isn't the case here. The arguments seem to refer to 'error' or 'uncertainty' — factors that harm the ability to differentiate between models (or 'noise'). So it may be better to say 'severe sample deficiencies'.

>>>The next section of the paper repeats the analyses of Sakamoto et al. This time they use a recent distribution of supertrees that includes approx 1000 species. They find that the group-wise model is strongly favoured, and includes a general tendency for rates to decrease through time, barring contrarian increases in some of the youngest subclades.

>>>See above, and comments in Bonsor et al. This model doesn't imply that dinosaurs 'lost their ability to speciate' (paraphrasing the conclusions of Sakamoto et al). But that is the conclusion fo the current work.

>"The results presented by Bonsor et al. [1], and claimed to refute our earlier analysis [2], do not offer any conclusive evidence. Their analyses suffer from: 1) an over-simplified model that is not comparable to our group-wise model; and 2) lack of statistical power owing to the extremely small sample sizes. We also demonstrate using 100 large trees (N = 961) of dinosaurs that the diversification decline model [2] is overwhelmingly supported when newer and better-sampled trees are used."

>>>I will be convinced by this once: (1) The authors provide clearer explanations and comparisons of effective sample size. (2) Ideally include a more well-reasoned explanation of their N = 50 sample size threshold. And (3) Acknowledge the other points made by Bonsor et al regarding (i) Model interpretation (that some groups show a contrarian pattern) and (ii) Theoretically non-sensible intercept terms.

Review form: Reviewer 2 (Martín Ezcurra)

Is the manuscript scientifically sound in its present form?

Yes

Are the interpretations and conclusions justified by the results?

Yes

Is the language acceptable?

Yes

Do you have any ethical concerns with this paper?

No

Have you any concerns about statistical analyses in this paper?

No

Recommendation?

Accept as is

Comments to the Author(s)

The manuscript by Sakamoto et al. provides a series of critics to the paper recently published by Bonsor et al. (2020) in this same journal. I consider that this is a healthy exercise of implementation of the scientific method. The reasons stated by Sakamoto et al. as critics to the Bonsor et al.'s analyses are soundly and overlying any statistically-based methodology. I would like to see a slightly more detailed discussion about the simulations conducted by previous authors about the statistical power of these analysis. Beyond that, I think that Sakamoto et al.'s

manuscript is worth of publishing after a few very minor changes indicated in the PDF version of the manuscript (see Appendix A).

Decision letter (RSOS-202143.R0)

Dear Dr Sakamoto

The Editors assigned to your paper RSOS-202143 "Overwhelming support for diversification decline in dinosaurs" have now received comments from reviewers and would like you to revise the paper in accordance with the reviewer comments and any comments from the Editors. Please note this decision does not guarantee eventual acceptance.

Note that the strongly critical reviewer, whose view has of course pushed us towards the requirement for major revision, sees your contribution as potentially worthwhile and publishable, but feels that your explanation/reasoning particularly with regard to the paper by Bonsor et al, could be much clearer and fairer. The other reviewer, who recommends publication, also recommends some strengthening of the discussion of previous work. So there is common ground between the two reviewers here.

Please submit your revised manuscript and required files (see below) no later than 21 days from today's (ie 06-May-2021) date. Note: the ScholarOne system will 'lock' if submission of the revision is attempted 21 or more days after the deadline. If you do not think you will be able to meet this deadline please contact the editorial office immediately.

on behalf of Peter Haynes (Subject Editor)
openscience@royalsociety.org

Associate Editor Comments to Author:

Comments to the Author:

Please accept the journal's apologies for the delay in completing review: while two reports had been received, the editors had been hoping that a third reviewer would return a report. In the event, they were unable to do so. Consequently, the decision has been based on two reviewers' reports. We hope you find their comments to be useful in preparing a revision.

Reviewer comments to Author:

Reviewer: 1

Comments to the Author(s)

I have read the comment article "Overwhelming Support for Diversification Decline in Dinosaurs" by Sakamoto et al. The article is a response to a previous paper by Bonsor et al, published last year. I also re-read that paper while conducting this review.

The comment by Sakamoto et al focuses on analysing data across the whole tree of dinosaurs, which includes hundreds of species. In contrast, Bonsor et al focussed on subtrees, and did not include variables to distinguish between the dynamics of diversification in different groups in their models. Sakamoto et al focus on those differences in approach to analysis. Here, Sakamoto et al argue that, once these things are done (large trees, clade-specific model terms), their original hypotheses findings are robustly supported.

They do not mention that Bonsor et al also levelled other criticisms of their original analysis. For example, their choice not to fix the intercept term to theoretically meaningful values. And the argument that the 'diversification decline', and arguments such as 'Late Cretaceous dinosaurs were incapable of speciation' (paraphrasing) do not take into account that some dinosaur subgroups were diversifying very rapidly in the Late Cretaceous, contrary to various statements in the original article by Sakamoto et al (e.g. dinosaurs lost their ability to speciate; paraphrasing). These other aspects seem to be integral to the criticisms of Bonsor et al but are not addressed in the response by Sakamoto et al. Importantly, the decision to not even acknowledge these other criticisms is being potentially misleading to readers of the Sakamoto et al response who may regard their summary as being a complete one. Acknowledging the multiple points of the arguments of Bonsor et al should therefore be a prerequisite of publication.

Sakamoto et al make two main points. The substance of one of these is that Bonsor et al did not use sufficient sample sizes, and did not allow clade-specific dynamics. However for some clades, the analyses of Bonsor et al include potentially similar sample sizes to those of subclades that were allowed to exhibit different diversification dynamics in the analyses of Sakamoto et al. I have provided some more information about this below. The overarching conclusion I have, based on the manuscript as currently written, is that the authors have presented an incomplete account of the nature of sample size, and have not provided sufficient information on effective sample sizes for a reader to really be able to evaluate their arguments. This needs to be addressed so that an informed reader would be able to make a fair comparison. Note, I am not saying that the authors are wrong. Indeed, I'm leaning towards the conclusion that they are right, at least so far as this specific set of model comparisons is concerned. However, there is important information missing that would allow me to fully judge the strength of their arguments.

Title: The title is over-blown and not strictly informative. Whether we consider a decline in diversification rates to be 'overwhelmingly supported' depends on factors other than those covered in the paper. For example, is the approach to testing this hypothesis appropriate in the first place? Is the assumption that nodes in a paleontological phylogeny provide unbiased information about counts of speciation events really justified (note: widespread methods such as the fossilize-birth death model assume this is -not- the case)? There has received essentially no scrutiny of method performance, so we don't really know the answer to these questions, and it's not really appropriate to get into the weeds on that topic in the present context. However, I'd like to see a more circumspect and less bombastic approach to writing, with more the characteristics of reflective science that acknowledges uncertainties. That is important here in the title and elsewhere in the manuscript.

In summary. The authors have a right to explain their position. But that explanation should acknowledge the multiple facets of the work by Bonsor et al rather than picking and choosing. Furthermore, the current manuscript should focus more on communicating information that is necessary for a reader to evaluate the strengths of the argument. And avoid bombastic language. Myself, I am part-way there. But I will be convinced by this once: (1) The authors provide clearer explanations and comparisons of effective sample size. (2) Ideally include a more well-reasoned explanation of their $N = 50$ sample size threshold. And (3) Acknowledge the other points made by Bonsor et al regarding (i) Model interpretation (that some groups show a contrarian pattern) and (ii) Theoretically non-sensible intercept terms. I view these aspects as being a relatively substantial re-tooling of the manuscript.

Detailed comments

>"Recently, Bonsor et al. [1] claimed to have put into question our preferred model of dinosaur diversification (number of speciation events through time) presented by Sakamoto et al. [2], by analysing an additional nine phylogenetic trees that were published subsequent to our study."

>>>This is not the entire substance of the arguments of Bonsor et al. Some aspects of the arguments are not even mentioned in the current article. The decision just to focus on one aspect allows the authors to potentially win that part of the argument. But it leaves other, potentially-important, aspects completely unaddressed. That's OK if they authors do not aim to convince people that they are correct. However, it is also potentially misleading because some readers may regard the summary provided here as being a complete account of the arguments of Bonsor et al.

>"which are not apparent from their article but evident from their data and codes: they used an overly simplified model formulation that is not comparable to our model"

>>>Ok, agreed.

>"and their analyses are generally based on small sample sizes."

>>>I agree with this in part. However, while reading the current ms I had various questions about the effective sample size of the analyses once terms are included to model different dynamics among subclades. The substance of my main question is as follows: Say I have a tree of 700 species, of which 40 are hadrosauroids. When I include a term to allow different dynamics of diversification in hadrosauroids, what is my effective sample size for inferring diversification dynamics in hadrosauroids? I believe it likely is $N = 40$. However it depends somewhat on model formulation because the model could treat their dynamics as inheriting some aspects from their

ancestry, or it could allow them to be completely decoupled. If complete decoupled, then the effective sample size for a subclade is likely to be the number of species sampled in that subclade (actually, a smaller number given the fact of hierarchical autocorrelation in phylogenetic comparative methods). Therefore, we need more information and the comparison needs to be more careful and overall transparent.

>>>The overarching conclusion I have, based on the manuscript as currently written, is that the authors have presented an incomplete account of the nature of sample size, and have not provided sufficient information on effective sample sizes for a reader to really be able to evaluate there arguments.

>"Sakamoto et al modelled separate effects of Time and Time2 on the individual subclades, Theropoda, Ornithischia, Sauropodomorpha, Ceratopsidae and Hadrosauriformes [2] within a single model framework. "

>>>Yeah, so basically the question here is whether I should expect different answers for sauropodomorph dynamics if I fit the model to just sauropodomorphs on their own? If the dynamics of sauropodomorph diversification in the 'whole Dinosauria' model are genuinely decoupled from those of other groups then the appropriate comparison of sample sizes would be between the count of sauropodomorphs included in each analysis. -Not-, as implied here, a comparison of the sample size of all Dinosauria in Sakamoto's original paper, to that for Sauropodomorpha in one of Bonsor et al's analyses. At least, that's where I stand on this topic at the moment. And if that is not the case, then it is the author's job to convince readers. This will require a much more reflective and circumspect approach than is currently manifested in the paper.

>"which, for some reason, is the model formulation that Bonsor et al. [1] chose to test instead"
>"...as Bonsor et al rightly point out (following our explicit and repeated statements...)"

>>>This language comes across as being pejorative and doesn't seem appropriate to me if the job is to convince using reasoned and compelling arguments.

>"When the group-wise model formulation is used... then the differences in DIC are indisputably in favour of the Time2 model, thus providing overwhelming support for this model of diversification through time"

>>>I see what the authors are trying to say here. However, the group-wise model also posits that some of the youngest groups (ceratopsian, hadrosauroids) show a different pattern that goes in the opposite direction. This is relevant to interpretation of this statistical model result. Namely, that the model might then -not- be interpreted as providing 'overwhelming' support for a diversification decline across all Dinosauria. This argument was made by Bonsor et al, and I feel that the authors here have failed to get to grips with it in an appropriately circumspect manner. See e.g. their title and various other overly-strong statements of unilateral interpretation.

>"The additional trees used in their analyses are very small (N = 23-141), with fewer species sampled per clade, compared to those we used (N = 420 and 614).

>>>The text starts 'species per clade', but the N values given for the original study of Sakamoto et al are not 'per clade values'. They are 'all Dinosauria' values. So, of course the numbers are large compared to the numbers used in the analyses of dinosaurs subclasses presented by Bonsor et al. However clearly this is a misleading as currently phrased. See my comment above about explanations of samples sizes and effective samples sizes. I feel it is likely, even after a 'fair' comparison, that the analyses of Sakamoto et al have larger effective sample sizes than the

analyses of Bonsor et al. Nevertheless, Sakamoto et al need to demonstrate this in a more transparent way.

>"In fact there is an even large meta tree..."

>>>That's true, it is large. Have the authors looked at the timescale phylogeny and which taxa are controlling the divergence times though? There are various species in there that have essentially unknown phylogenetic affinities (based on comparative anatomical and primary phylogenetic literature). Those taxa sometimes are placed in phylogenetic positions that are deeply nested compared to what you'd guess based on their times of appearance. This has a big effect on the divergence times found by Lloyd et al. Sure, OK, it's a tree that exists and could be analysed. But the fact that many of the divergence times are pushed to surprisingly old ages is probably artifactual and I believe it would tend to bias in analysis in favour of the hypothesis of Sakamoto et al.

>"Larger sample sizes offer more statistical power in estimating parameters in complex models of diversification. Conversely, smaller sample sizes lack the appropriate statistical power to differentiate between competing models. This is especially so for trees with $N < 50$. Simulations have demonstrated that simple evolutionary parameters like heritability"

>>>Sure, I agree with the principle that large sample sizes give more statistical power. However, the text is very generic. It asserts that variation in sample size within the range of the analyses of Bonsor et al will be important in the specific example. But doesn't provide evidence from simulations that are relevant to the current estimation task. Instead, it refers to estimating heritability, which is really quite a different inference problem. It also lacks nuance. For example, greater statistical power is particularly important when effect sizes are small. So if there is 'overwhelming' support (i.e. large effect sizes), then we may not expect this to be as much of a problem. Note that I'm not saying that the authors are wrong. Instead, I'm saying that the number 'N=50' seems to have been chosen arbitrarily without much rationale behind it. I don't think this way of making the argument sets a good example for others to follow.

>>>Also, what is the effective sample size for this type of analysis. Cecile Ané did some nice work showing the effective sample sizes for phylogenetic comparative methods could be much lower than the number of species samples, and that the effective sample size increases initially rapidly when few species have been sampled, and then slows down, with new additions of species adding relatively little new information. Of course, she was looking at a different estimation problem. I only write this here to indicate that it isn't possible to say a priori what sample size is needed without further evaluation of method performance. That evaluation has not been done, to my knowledge, but could provide some of the best support for the authors' preferred hypothesis.

>"In fact, Bonsor and colleagues [1] could not differentiate between the null model and the two competing models in the majority of the trees they analysed, indicating that their sample sizes do not offer sufficient statistical power".

>>>There's no way to approach this other than to say that this reasoning is clearly flawed from a the most basic level of statistics. The fact that the null hypothesis cannot be rejected could -either- reflect low statistical power, as argued here, -or- that there is no evidence for the alternative hypothesis. Please remove or re-phrase this specific sentence.

>"Additionally, counting the number of nodes on smaller trees inevitably leads to a narrower range of speciation events (Fig. 1)"

>>>The fact that smaller trees have fewer nodes is self-evident. Please replace 'narrower range' with 'smaller count' or similar.

>"...at least three of their trees super from a lack of variation on the main predictor variable, Time'

>>>Agreed.

>"Thus, arguing that one model is not discernible from another, or indeed even from conclusive. In fact, such results suffer from severe sampling biases".

>>>'Bias' refers to systematic effects that would incorrectly prefer one model over others. That isn't the case here. The arguments seem to refer to 'error' or 'uncertainty' – factors that harm the ability to differentiate between models (or 'noise'). So it may be better to say 'severe sample deficiencies'.

>>>The next section of the paper repeats the analyses of Sakamoto et al. This time they use a recent distribution of supertrees that includes approx 1000 species. They find that the group-wise model is strongly favoured, and includes a general tendency for rates to decrease through time, barring contrarian increases in some of the youngest subclades.

>>>See above, and comments in Bonsor et al. This model doesn't imply that dinosaurs 'lost their ability to speciate' (paraphrasing the conclusions of Sakamoto et al). But that is the conclusion fo the current work.

>"The results presented by Bonsor et al. [1], and claimed to refute our earlier analysis [2], do not offer any conclusive evidence. Their analyses suffer from: 1) an over-simplified model that is not comparable to our group-wise model; and 2) lack of statistical power owing to the extremely small sample sizes. We also demonstrate using 100 large trees (N = 961) of dinosaurs that the diversification decline model [2] is overwhelmingly supported when newer and better-sampled trees are used."

>>>I will be convinced by this once: (1) The authors provide clearer explanations and comparisons of effective sample size. (2) Ideally include a more well-reasoned explanation of their N = 50 sample size threshold. And (3) Acknowledge the other points made by Bonsor et al regarding (i) Model interpretation (that some groups show a contrarian pattern) and (ii) Theoretically non-sensible intercept terms.

Reviewer: 2

Comments to the Author(s)

The manuscript by Sakamoto et al. provides a series of critics to the paper recently published by Bonsor et al. (2020) in this same journal. I consider that this is a healthy exercise of implementation of the scientific method. The reasons stated by Sakamoto et al. as critics to the Bonsor et al.'s analyses are soundly and overlying any statistically-based methodology. I would like to see a slightly more detailed discussion about the simulations conducted by previous authors about the statistical power of these analysis. Beyond that, I think that Sakamoto et al.'s manuscript is worth of publishing after a few very minor changes indicated in the PDF version of the manuscript.

===PREPARING YOUR MANUSCRIPT===

===PREPARING YOUR REVISION IN SCHOLARONE===

- An individual file of each figure (EPS or print-quality PDF preferred [either format should be produced directly from original creation package], or original software format).
 - An editable file of each table (.doc, .docx, .xls, .xlsx, or .csv).
 - An editable file of all figure and table captions.
- Note: you may upload the figure, table, and caption files in a single Zip folder.
- Any electronic supplementary material (ESM).
 - If you are requesting a discretionary waiver for the article processing charge, the waiver form must be included at this step.
 - If you are providing image files for potential cover images, please upload these at this step, and inform the editorial office you have done so. You must hold the copyright to any image provided.
 - A copy of your point-by-point response to referees and Editors. This will expedite the preparation of your proof.

- Ensure that your data access statement meets the requirements at <https://royalsociety.org/journals/authors/author-guidelines/#data>. You should ensure that you cite the dataset in your reference list. If you have deposited data etc in the Dryad repository, please include both the 'For publication' link and 'For review' link at this stage.
- If you are requesting an article processing charge waiver, you must select the relevant waiver option (if requesting a discretionary waiver, the form should have been uploaded at Step 3 'File upload' above).
- If you have uploaded ESM files, please ensure you follow the guidance at <https://royalsociety.org/journals/authors/author-guidelines/#supplementary-material> to include a suitable title and informative caption. An example of appropriate titling and captioning may be found at https://figshare.com/articles/Table_S2_from_Is_there_a_trade-off_between_peak_performance_and_performance_breadth_across_temperatures_for_aerobic_sc_ope_in_teleost_fishes_/3843624.

Author's Response to Decision Letter for (RSOS-202143.R0)

See Appendix B.

Decision letter (RSOS-202143.R1)

Dear Dr Sakamoto,

I am pleased to inform you that your manuscript entitled "Strong support for a heterogeneous speciation decline model in Dinosauria: Refuting false claims made by Bonsor et al. (2020)" is now accepted for publication in Royal Society Open Science.

on behalf of Peter Haynes (Subject Editor)
openscience@royalsociety.org

Appendix A**ROYAL SOCIETY
OPEN SCIENCE****Overwhelming support for diversification decline in
dinosaurs**

Journal:	Royal Society Open Science
Manuscript ID	RSOS-202143
Article Type:	Comment
Date Submitted by the Author:	25-Nov-2020
Complete List of Authors:	Sakamoto, Manabu; University of Lincoln, School of Life Sciences Benton, Michael; University of Bristol, Earth Sciences Venditti, Chris; University of Reading, School of Biological Sciences
Subject:	palaeontology < BIOLOGY, evolution < BIOLOGY
Keywords:	dinosaurs, phylogenetic comparative methods, GLMM, diversification rate, diversification decline
Subject Category:	Earth and Environmental Science

Overwhelming support for diversification decline in dinosaurs

Manabu Sakamoto¹, Mike Benton² and Chris Venditti³

¹School of Life Sciences, University of Lincoln, Lincoln, UK

²School of Earth Sciences, University of Bristol, Bristol, UK

³School of Biological Sciences, University of Reading, Reading, UK

Recently, Bonsor et al. [1] claimed to have put into question our preferred model of dinosaur diversification (number of speciation events through time) presented by Sakamoto et al. [2], by analysing ~~an additional nine~~ phylogenetic trees that were published subsequent to our study. However, their data and analyses are not sufficient to substantiate such a claim for two reasons, which are not apparent from their article but evident from their data and codes: they used an overly simplified model formulation that is not comparable to our model; and their analyses are generally based on small sample sizes.

Global vs group-wise model

Sakamoto et al. modelled separate effects of Time and  Time on the individual subclades, Theropoda, Ornithischia, Sauropodomorpha, Ceratopsidae and Hadrosauriformes [2] within a single model framework. This model outcompeted all other alternative models, including the global Dinosauria-wide model (single set of parameters across the entire tree), which, for some reason, is the model formulation that Bonsor et al. [1] chose to test instead. If the global Dinosauria-wide model was used to compare between the Time² and $\sqrt{\text{Time}}$ models, then it is indeed the case that the DIC difference between models is smaller and

1
2
3 often not significant under the cut-off value of 4 [2]. However, as Bonsor et al. [1] rightly
4
5 point out (following our explicit and repeated statements [2]), patterns of diversification are
6
7 not homogenous across the board. Indeed, we demonstrated [2] that a group-wise model
8
9 was overwhelmingly supported through DIC comparisons. We explicitly modelled this
10
11 heterogeneity across the various clades and discussed potential evolutionary mechanisms
12
13 distinguishing diversification patterns in Ceratopsidae and Hadrosauriformes from the other
14
15 clades. When the group-wise model formulation is used to compare between the Time² and
16
17 $\sqrt{\text{Time}}$ models, then the differences in DIC are indisputably in favour of the Time² model,
18
19 thus providing *overwhelming support* for this model of diversification through time [2].
20
21
22
23
24
25
26
27
28

29 Taxonomic coverage and range of data

30
31
32
33 While Bonsor et al. [1] claim that the phylogenetic trees we used [2] have been
34
35 superseded by trees that have been published since, the additional trees used in their
36
37 analyses are very small ($N = 23\text{--}141$; Fig. 1), with fewer species sampled per clade,
38
39 compared to those we used ($N = 420$ and 614). In fact, there is an even larger meta tree ($N =$
40
41 961) available [3] that was not used by Bonsor et al. [1] (analysed here for completeness).
42
43 Larger sample sizes offer more statistical power in estimating parameters in complex
44
45 models of diversification. Conversely, smaller sample sizes lack the appropriate statistical
46
47 power to differentiate between competing models. This is especially so for trees with $N <$
48
49 50 . Simulations have demonstrated that simple evolutionary parameters like heritability H^2
50
51 (also known as λ) [4] cannot be reliably estimated at such smaller sample sizes [5,6]. This
52
53 will severely affect models of diversification such as those modelled by Sakamoto et al. [2]
54
55
56
57
58
59
60

and Bonsor et al. [1]. In fact, Bonsor and colleagues [1] could not differentiate between the null model and the two competing models in the majority of the trees they analysed, indicating that their sample sizes do not offer sufficient statistical power. The null model was consistently and definitively rejected when sample sizes are sufficiently large [2].

Figure 1. Node count is plotted against Time (here expressed as the last appearance dates in Myr ago) for the meta-tree of Dinosauria ($N = 961$) [3] in grey. Node count and Time from each of the nine trees used by [1] are superimposed in maroon. Eight out of the nine trees have sample sizes less than 100 with three less than 50. The range of node counts covered by the nine trees are far narrower than those taken from the $N = 961$ meta-tree. The span in Time is also limited. Such narrow ranges in both

1
2
3 the response and predictor variables, coupled with a lack of statistical power owing to small sample
4 sizes, lead to inconclusive model selection.
5
6

7 Additionally, counting the number of nodes on smaller trees inevitably leads to a
8 narrower range of speciation events (Fig. 1). The effects of under-sampling are clearly
9 evident in their Node Count variable compared to those taken from a larger, more
10 comprehensively sampled tree (Fig. 1). Furthermore, at least three of their trees suffer from
11 a lack of variation on the main predictor variable, Time (Fig. 1). With such limited ranges in
12 the Time variable, Node Count rarely comes under any significant effects from Time^2 or
13 $\sqrt{\text{Time}}$. This is another reason why Bonsor et al. [1] could not reject the null model.
14
15
16
17
18
19
20
21
22
23

24 Thus, arguing that one model is not discernible from another, or indeed even from
25 the null model, based on results from poorly sampled and narrow-ranged data, is far from
26 conclusive. In fact, such results suffer from severe sampling biases.
27
28
29
30
31
32
33
34

35 Reanalysis of diversification through time in dinosaurs over a 36 sample of giant trees 37 38 39 40 41 42 43

44 For completeness, we modelled diversification through time on a sample of recent
45 meta-trees of Dinosauria [3], which covers 961 species. We selected 100 topologies at
46 random from the sample of 1000 trees provided in the SI [3]. We scaled the branches to
47 cover both extremes of branch lengths, the shortest possible tree and the longest possible
48 tree. We analysed both sets of trees (i.e. 200 trees in total).
49
50
51
52
53
54
55

56 We fitted three models for each tree topology: the null linear model (Time model;
57 Node Count \sim Time); the quadratic model (Time^2 model; Node Count \sim Time + Time^2); and
58
59
60

1
2
3 the square root model ($\sqrt{\text{Time}}$ model; Node Count $\sim \sqrt{\text{Time}}$). We estimated separate model
4
5 parameters for each of the five subclades, Ceratopsidae, Hadrosauriformes, Ornithischia
6
7 (non-ceratopsid, non-hadrosaurid), Sauropodomorpha, and Theropoda. For Ceratopsidae
8
9 and Hadrosauriformes, Time^2 and $\sqrt{\text{Time}}$ were not modelled and instead only the Time
10
11 effects were modelled. We compared model fit using DIC with a threshold difference value
12
13 of 4.
14
15
16
17

18 In both sets of 100 trees analysed, the difference in DIC between the Time^2 and
19
20 $\sqrt{\text{Time}}$ models was at least 9 and 10 for the shortest and longest trees respectively (with the
21
22 median being 13 and 15 respectively) in favour of the Time^2 model. The null model was
23
24 firmly rejected in every comparison. Therefore, we provide here yet again, overwhelming
25
26 support for the Time^2 model over the $\sqrt{\text{Time}}$ or the null models, as we showed before [2].
27
28
29
30
31
32
33

34 Conclusion

35
36
37
38 The results presented by Bonsor et al. [1], and claimed to refute our earlier analysis
39
40 [2], do not offer any conclusive evidence. Their analyses suffer from: 1) an over-simplified
41
42 model that is not comparable to our group-wise model; and 2) lack of statistical power
43
44 owing to the extremely small sample sizes. We also demonstrate using 100 large trees ($N =$
45
46 961) of dinosaurs that the diversification decline model [2] is overwhelmingly supported
47
48 when newer and better-sampled trees are used.
49
50
51
52
53
54
55
56
57
58
59
60

References

1. Bonsor JA, Barrett PM, Raven TJ, Cooper N. In press. Dinosaur diversification rates were not in decline prior to the K-Pg boundary. *Royal Society Open Science* **7**, 201195. (doi:10.1098/rsos.201195)
2. Sakamoto M, Benton MJ, Venditti C. 2016 Dinosaurs in decline tens of millions of years before their final extinction. *Proceedings of the National Academy of Sciences* **113**, 5036–5040. (doi:10.1073/pnas.1521478113)
3. Lloyd GT, Bapst DW, Friedman M, Davis KE. 2016 Probabilistic divergence time estimation without branch lengths: dating the origins of dinosaurs, avian flight and crown birds. *Biology Letters* **12**.
4. Pagel M. 1997 Inferring evolutionary processes from phylogenies. *Zoologica Scripta* **26**, 331–348. (doi:10.1111/j.1463-6409.1997.tb00423.x)
5. Freckleton RP, Harvey PH, Pagel M. 2002 Phylogenetic analysis and comparative data: A test and review of evidence. *American Naturalist* **160**, 712--726.
6. Sakamoto M, Venditti C. 2018 Phylogenetic non-independence in rates of trait evolution. *Biology Letters* **14**, 20180502. (doi:10.1098/rsbl.2018.0502)

Appendix B

Dear Anita Kristiansen,

Please find below our response to reviewer comments. We address every point made by the reviewers, in particular, those made by Reviewer 1. Please note, much of Reviewer 1's criticisms are based on misunderstandings of our original paper from 2016 as well as taking the erroneous criticisms made by Bonsor et al. at face value. We believe this reviewer is compromised and biased in favour of Bonsor et al. We have made efforts to address their extensive criticisms here and in the revised manuscript. In doing so we believe we have strengthened our arguments.

As this process has taken an unexpectedly long time, we would very much appreciate it if this revision process would be treated with high priority.

Reviewer comments are in blue while our responses are in black.

Yours sincerely,

Manabu Sakamoto, Mike Benton, and Chris Venditti

Dear Dr Sakamoto

The Editors assigned to your paper RSOS-202143 "Overwhelming support for diversification decline in dinosaurs" have now received comments from reviewers and would like you to revise the paper in accordance with the reviewer comments and any comments from the Editors. Please note this decision does not guarantee eventual acceptance.

We have fully taken the reviewer comments on board and fundamentally revised our manuscript. We take Reviewer 1's recommendation and address each criticism made by Bonsor et al. point by point.

Note that the strongly critical reviewer, whose view has of course pushed us towards the requirement for major revision, sees your contribution as potentially worthwhile and publishable, but feels that your explanation/reasoning particularly with regard to the paper by Bonsor et al, could be much clearer and fairer. The other reviewer, who recommends

publication, also recommends some strengthening of the discussion of previous work. So there is common ground between the two reviewers here.

Please submit your revised manuscript and required files (see below) no later than 21 days from today's (ie 06-May-2021) date. Note: the ScholarOne system will 'lock' if submission of the revision is attempted 21 or more days after the deadline. If you do not think you will be able to meet this deadline please contact the editorial office immediately.

Kind regards,

Anita Kristiansen
Editorial Coordinator

on behalf of Peter Haynes (Subject Editor)
openscience@royalsociety.org

Associate Editor Comments to Author:

Comments to the Author:

Please accept the journal's apologies for the delay in completing review: while two reports had been received, the editors had been hoping that a third reviewer would return a report. In the event, they were unable to do so. Consequently, the decision has been based on two reviewers' reports. We hope you find their comments to be useful in preparing a revision.

Reviewer comments to Author:

Reviewer: 1

Comments to the Author(s)

I have read the comment article "Overwhelming Support for Diversification Decline in Dinosaurs" by Sakamoto et al. The article is a response to a previous paper by Bonsor et al, published last year. I also re-read that paper while conducting this review.

The comment by Sakamoto et al focuses on analysing data across the whole tree of dinosaurs, which includes hundreds of species. In contrast, Bonsor et al focussed on

subtrees, and did not include variables to distinguish between the dynamics of diversification in different groups in their models. Sakamoto et al focus on those differences in approach to analysis. Here, Sakamoto et al argue that, once these things are done (large trees, clade-specific model terms), their original hypotheses findings are robustly supported.

They do not mention that Bonsor et al also levelled other criticisms of their original analysis. For example, their choice not to fix the intercept term to theoretically meaningful values. We had included a discussion on this point in an initial draft, but we decided to omit this from the submitted manuscript, owing to the tight word count restrictions (<https://royalsocietypublishing.org/rsos/for-authors#question1>) (please see screenshot below).

Comment and invited reply

Comments are self-proposed by any reader shortly after the initial article is published, and a reply will be submitted by the original research authors as a response. Please refer to our comment and reply policy page for more information about these article types, or please contact the Editorial Office prior to submission with any queries. The word limit for comments and replies is 1000 words.

However, this point is discussed in Sakamoto et al. (2016), in which we outline and justify our reasons for including an intercept term in our model. We repeat here for the benefit of the reviewer, but an intercept fixed at zero would imply that we know for certain that node count was zero at the estimated root age of the tree. However, we acknowledge that this is highly unlikely and that in the case that the root was older than estimated for our trees, then we would be forcing a model in which an unknown number of basal divergences are ignored. The intercept acts as an offset to account for this uncertainty over missing nodes. The following screen captures are from the SI of our 2016 paper (<https://www.pnas.org/content/113/18/5036>):

The number of nodes at time 0 is zero, and thus, the model should theoretically be fitted through the origin. However, our model is more conservative, in that we allow intercepts to be estimated; by estimating the intercept, we account for potential uncertainties in the basal divergences (e.g., missing taxa or incorrect dating of the root).

The estimated intercept models consistently had significantly better model fit (lower DIC scores) than those with intercepts forced at zero ($\Delta\text{DIC} > 100$). This result indicates that we are predicting more speciation events at the base of the tree than would be expected given the amount of time elapsed since the root. Either the fossil record of the earliest dinosaurs is incomplete (which obviously, is very likely) or our root estimate is too young (which according to a recent study dating the origin of dinosaurs to ~236–234 Mya, a date considerably younger than the root constraint that we use here, would be very unlikely); thus, by estimating intercepts, we allow for these uncertainties in the model. However, we get qualitatively identical results when intercepts are fixed to zero as those when intercepts are estimated.

Although we mentioned in the 2016 paper as shown in the screen shot above that the results are qualitatively identical between models where the intercept is estimated or where it is fixed at zero, the plotted predictions for the zero-intercept model shows stronger downturn effects than that for the model where the intercept is estimated. Thus, a model in which the intercept is estimated is more conservative than a zero-intercept model, both in terms of allowing for missing basal divergences but also in the downturn effect.

Having said that, we have substantially reworked our manuscript to address these additional points.

And the argument that the 'diversification decline', and arguments such as 'Late Cretaceous dinosaurs were incapable of speciation' (paraphrasing) do not take into account that some dinosaur subgroups were diversifying very rapidly in the Late Cretaceous, contrary to various statements in the original article by Sakamoto et al (e.g. dinosaurs lost their ability to speciate; paraphrasing).

We say this with all due respect, but we feel part of the problem with this discourse is that disagreements and arguments all stem from paraphrased 'quotes'. The wording we used was not as definitive as 'Late Cretaceous dinosaurs were incapable of speciation' or 'dinosaurs lost their ability to speciate'. The precise wording we used are shown in the screen-captures below. We only count a further two instances where we mention a 'reduction in their capacity to replace extinct species'. In light of the reviewer's concerns over our choice in wording in our current manuscript, we believe precise quoting of sources (not paraphrasing) is critical.

Significance

Whether dinosaurs were in decline before their final extinction 66 Mya has been debated for decades with no clear resolution. This dispute has not been resolved because of inappropriate data and methods. Here, for the first time to our knowledge, we apply a statistical approach that models changes in speciation and extinction through time. We find overwhelming support for a long-term decline across all dinosaurs and within all three major dinosaur groups. Our results highlight that dinosaurs showed a marked reduction in their ability to replace extinct species with new ones, making them vulnerable to extinction and unable to respond quickly to and recover from the final catastrophic event 66 Mya.

Abstract

Whether dinosaurs were in a long-term decline or whether they were reigning strong right up to their final disappearance at the Cretaceous–Paleogene (K-Pg) mass extinction event 66 Mya has been debated for decades with no clear resolution. The dispute has continued unresolved because of a lack of statistical rigor and appropriate evolutionary framework. Here, for the first time to our knowledge, we apply a Bayesian phylogenetic approach to model the evolutionary dynamics of speciation and extinction through time in Mesozoic dinosaurs, properly taking account of previously ignored statistical violations. We find overwhelming support for a long-term decline across all dinosaurs and within all three dinosaurian subclades (Ornithischia, Sauropodomorpha, and Theropoda), where speciation rate slowed down through time and was ultimately exceeded by extinction rate tens of millions of years before the K-Pg boundary. The only exceptions to this general pattern are the morphologically specialized herbivores, the Hadrosauriformes and Ceratopsidae, which show rapid species proliferations throughout the Late Cretaceous instead. Our results highlight that, despite some heterogeneity in speciation dynamics, dinosaurs showed a marked reduction in their ability to replace extinct species with new ones, making them vulnerable to extinction and unable to respond quickly to and recover from the final catastrophic event.

With regard to the reviewer’s criticism that we “do not take into account that some dinosaur subgroups were diversifying very rapidly in the Late Cretaceous, contrary to various statements in the original article by Sakamoto et al”, we refer directly to our 2016 paper again. The screenshot below is from the abstract:

tens of millions of years before the K-Pg boundary. The only exceptions to this general pattern are the morphologically specialized herbivores, the Hadrosauriformes and Ceratopsidae, which show rapid species proliferations throughout the Late Cretaceous instead. Our results highlight that, despite some heterogeneity in speciation dynamics,

Similarly, below is from the main text:

However, these two subclades combined only represent 14% of dinosaur species; over time, dinosaurs overwhelmingly experienced a reduction in their capacity to replace extinct species with new ones—net speciation per 1 My at the time that dinosaurs went extinct (66 Mya) was significantly below zero (speciation rate is less than extinction rate) (**Fig. 3B**) in the three major clades (**Table S12**)—and Hadrosauriformes and Ceratopsidae are the exceptions.

The heterogeneity in diversification dynamics was a major point of discussion in our 2016 paper, even appearing in our abstract (please see screenshot above) as well as being modelled as parameters in our selected model, the 5-group quadratic model. As evident from the screenshot above, our statement that we find ‘overwhelming support for a diversification decline’ is immediately followed by a qualifier, ‘the only exceptions to this general pattern are the [...] Hadrosauriformes and Ceratopsidae [...]’. In fact, the very

statement (the one which the reviewer paraphrased) appears in a sentence in the abstract following the qualifier 'despite some heterogeneity in speciation dynamics' (again, please see the screenshot above), clearly acknowledging the presence of heterogeneous diversification dynamics across the dinosaur tree. Paraphrased statements such as that made by the reviewer take our original statement out of context and are thus misleading.

These other aspects seem to be integral to the criticisms of Bonsor et al but are not addressed in the response by Sakamoto et al. Importantly, the decision to not even acknowledge these other criticisms is being potentially misleading to readers of the Sakamoto et al response who may regard their summary as being a complete one. Acknowledging the multiple points of the arguments of Bonsor et al should therefore should be a prerequisite of publication.

We were restricted by the 1000-word limit as outlined above. In the revised manuscript we offer a point-by-point rebuttal of every erroneous claim made by Bonsor et al. It is unfortunate and sad that this treatment is necessary, but we heed the reviewer's suggestions in a truly reflective way.

Sakamoto et al make two main points. The substance of one of these is that Bonsor et al did not use sufficient sample sizes, and did not allow clade-specific dynamics. However for some clades, the analyses of Bonsor et al include potentially similar sample sizes to those of subclades that were allowed to exhibit different diversification dynamics in the analyses of Sakamoto et al. I have provided some more information about this below. The overarching conclusion I have, based on the manuscript as currently written, is that the authors have presented an incomplete account of the nature of sample size, and have not provided sufficient information on effective sample sizes for a reader to really be able to evaluate their arguments. This needs to be addressed so that an informed reader would be able to make a fair comparison. Note, I am not saying that the authors are wrong. Indeed, I'm leaning towards the conclusion that they are right, at least so far as this specific set of model comparisons is concerned. However, there is important information missing that would allow me to fully judge the strength of their arguments.

The selection amongst the three models, the time-linear, time-square-root and time-quadratic models, in both Bonsor et al. and our 2016 paper, is based on DIC comparisons, not based on the group-wise parameters. The DIC value is conditioned on the tree and data that the model was fitted on, so for a 5-group quadratic model, the 'effective' sample size (that matters for model selection) would be that of the entire dataset, not subclades. Since the central tests conducted by Bonsor et al. involve model selection by DIC comparisons, what matters is the sample size of the DIC estimation, not the effective sample sizes of the group-wise parameters. We emphasise again that the tests conducted by Bonsor et al. on their nine trees are based on DIC comparisons. This is very important and key to understanding why the tree-wide sample size (the sample on which the model was fitted) is comparable across the different datasets, not equivalently-sized subclades.

Title: The title is over-blown and not strictly informative. Whether we consider a decline in diversification rates to be 'overwhelmingly supported' depends on factors other than those covered in the paper. For example, is the approach to testing this hypothesis appropriate in the first place? Is the assumption that nodes in a paleontological phylogeny provide unbiased information about counts of speciation events really justified (note: widespread

methods such as the fossilize-birth death model assume this is -not- the case)? There has received essentially no scrutiny of method performance, so we don't really know the answer to these questions, and it's not really appropriate to get into the weeds on that topic in the present context. However, I'd like to see a more circumspect and less bombastic approach to writing, with more the characteristics of reflective science that acknowledges uncertainties. That is important here in the title and elsewhere in the manuscript.

We have changed the title. We hope the reviewer is satisfied.

In summary. The authors have a right to explain their position. But that explanation should acknowledge the multiple facets of the work by Bonsor et al rather than picking and choosing.

We mentioned this above already, but we repeat here for completeness: we were restricted by the 1000-word limit for Comments as outlined in the guidelines to authors. However, we have provided a point-by-point rebuttal of all claims made by Bonsor et al. We reiterate here again that most of the criticisms raised by Bonsor et al. are misleading and unjustified given that we had addressed these in our 2016 paper already. That Bonsor et al. failed to acknowledge this is the core of the problem.

Furthermore, the current manuscript should focus more on communicating information that is necessary for a reader to evaluate the strengths of the argument. And avoid bombastic language.

It is puzzling why misleading false claims, false equivalency and other logical fallacies in Bonsor et al.'s work is fine but our use of 'overwhelming support' in our response to their criticism is not.

Myself, I am part-way there. But I will be convinced by this once:

(1) The authors provide clearer explanations and comparisons of effective sample size.

We no longer discuss the effects of sample size and have refocused our criticism on falsely equating dynamics at the sub-clade level with those at a larger clade level e.g., ankylosaurs (Arbour dataset) as equivalent of ornithischians, or coelurosaurs (Cau et al. dataset) as equivalent of theropods. The Cau et al. dataset only has two non-coelurosaurian allosauroids as outgroup while the rest are all coelurosaurs (139 out of 141). This is clearly a false equivalency. The nine sub-clade level trees are not equivalent to the near-whole clades (Ornithischia, Sauropodomorpha, Theropoda) tested in our models.

(2) Ideally include a more well-reasoned explanation of their $N = 50$ sample size threshold.

This is not really a threshold as such but something that is born out of simulations from previous research. We explain and cite this in our manuscript:

50	
51	power to differentiate between competing models. This is especially so for trees with $N <$
52	
53	50. Simulations have demonstrated that simple evolutionary parameters like heritability H^2
54	
55	(also known as λ) [4] cannot be reliably estimated at such smaller sample sizes [5,6]. This
56	

However, this is largely irrelevant in the revised manuscript as we no longer focus on sample size itself.

And (3) Acknowledge the other points made by Bonsor et al regarding (i) Model interpretation (that some groups show a contrarian pattern)

As outlined above, we discussed this in our original 2016 paper, and we did not see a reason to repeat this again. However, we revised the manuscript to clarify that Bonsor et al.'s criticism that we did not acknowledge heterogenous diversification dynamics is false and that we did indeed take this into account. It is, in fact, stated clearly and upfront in the abstract (please see screenshot above).

and (ii) Theoretically non-sensible intercept terms.

Please see our response above with respect to intercept terms. We have tested zero-intercept models in our 2016 paper. We have included this in our revised manuscript. For completeness we repeat here that this is to account for uncertainties surrounding potential missing nodes at the base of the tree but also because DIC comparisons clearly would favour the model where the intercept is estimated over the zero-intercept model.

I view these aspects as being a relatively substantial re-tooling of the manuscript. It is a substantial reworking of the manuscript.

Detailed comments

>"Recently, Bonsor et al. [1] claimed to have put into question our preferred model of dinosaur diversification (number of speciation events through time) presented by Sakamoto et al. [2], by analysing an additional nine phylogenetic trees that were published subsequent to our study."

>>>This is not the entire substance of the arguments of Bonsor et al. Some aspects of the arguments are not even mentioned in the current article. The decision just to focus on one aspect allows the authors to potentially win that part of the argument. But it leaves other, potentially-important, aspects completely unaddressed. That's OK if they authors do not aim to convince people that they are correct. However, it is also potentially misleading because some readers may regarding the summary provided here as being a complete account of the arguments of Bonsor et al.

We repeat here that we were under tight word restrictions. We have revised our manuscript to explicitly refute each of Bonsor et al.'s erroneous criticisms.

>"which are not apparent from their article but evident from their data and codes: they used an overly simplified model formulation that is not comparable to our model"

>>>Ok, agreed.

>"and their analyses are generally based on small sample sizes."

>>>I agree with this in part. However, while reading the current ms I had various questions about the effective sample size of the analyses once terms are included to model different dynamics among subclades. The substance of my main question is as follows: Say I have a tree of 700 species, of which 40 are hadrosauroids. When I include a term to allow different dynamics of diversification in hadrosauroids, what is my effective sample size for inferring

diversification dynamics in hadrosauroids? I believe it likely is $N = 40$. However it depends somewhat on model formulation because the model could treat their dynamics as inheriting some aspects from their ancestry, or it could allow them to be completely decoupled. If complete decoupled, then the effective sample size for a subclade is likely to be the number of species sampled in that subclade (actually, a smaller number given the fact of hierarchical autocorrelation in phylogenetic comparative methods). Therefore, we need more information and the comparison needs to be more careful and overall transparent.

We no longer discuss the effects of sample size in our revised manuscript. Instead, we have refocused our criticism on falsely equating dynamics at the sub-clade level with those at a larger clade level e.g., ankylosaurs (Arbour dataset) as equivalent of ornithischians, or coelurosaurs (Cau et al. dataset) as equivalent of theropods. The Cau et al. dataset only has two non-coelurosaurian allosauroids as outgroup while the rest are all coelurosaurs (139 out of 141). This is clearly a false equivalency. The nine sub-clade level trees are not equivalent to the near-whole clades (Ornithischia, Sauropodomorpha, Theropoda) tested in our models. This is the same as taking ceratopsids or hadrosauriforms and equating it to Ornithischia. There will be a discrepancy in the best-fit models and the conclusion will be 'uncertain' but this uncertainty only exists because diversification dynamics are compared across widely different subclades with the larger more inclusive clade as if they are all equivalent.

Having said that, we address here the reviewer's question regarding the effective sample size in an example case for hadrosauriforms. The model selection routine that we and Bonsor et al. employed depends on DIC values. That is the best model is selected based on the lowest DIC score and preferably one with the difference >4 . If the quadratic model had the lowest DIC score, then we would select the quadratic model and interpret diversification dynamic accordingly. Thus, the unique diversification dynamics detected for hadrosauriforms in our 5-group quadratic model was selected based on the DIC score computed for that model on the entire dataset. Thus, the effective sample size to determine the diversification dynamics was the sample size for the entire dataset and tree. If the same diversification dynamics were to be detected on a hadrosauriform tree, then the effective sample size for this set would be the sample size of the hadrosauriform tree.

>>>The overarching conclusion I have, based on the manuscript as currently written, is that the authors have presented an incomplete account of the nature of sample size, and have not provided sufficient information on effective sample sizes for a reader to really be able to evaluate their arguments.

Please see above for our revised argument about small/sparse sample subtrees and the false equivalency of comparing these against more inclusive clades.

>"Sakamoto et al modelled separate effects of Time and Time² on the individual subclades, Theropoda, Ornithischia, Sauropodomorpha, Ceratopsidae and Hadrosauriformes [2] within a single model framework. "

>>>Yeah, so basically the question here is whether I should expect different answers for sauropodomorph dynamics if I fit the model to just sauropodomorphs on their own? If the dynamics of sauropodomorph diversification in the 'whole Dinosauria' model are genuinely decoupled from those of other groups then the appropriate comparison of sample sizes

would be between the count of sauropodomorphs included in each analysis. -Not-, as implied here, a comparison of the sample size of all Dinosauria in Sakamoto's original paper, to that for Sauropodomorpha in one of Bonsor et al's analyses. At least, that's where I stand on this topic at the moment.

This is statistically incorrect. The likelihood of the model and thereby the DIC are based on the entire tree and dataset, not subsets (i.e., per subclade). The group-wise models from Sakamoto et al. (2016) as well as our additional analyses here only return one DIC value per tree, not per clade. Our models were selected on the basis of these tree-wide DIC values. That is, the diversification dynamics are selected based on DIC values, not on the parameters themselves, so the sample size that matters is the one that the DIC is based on, which is the total sample size of the dataset involved. Thus, a fair comparison or critique should be made on a Dinosauria-level model, which is what we did in our current manuscript using a substantially larger tree (N=961).

On the point of a comparison between model parameters estimated on a Sauropodomorpha tree (the Carballido tree analysed by Bonsor et al.) and those estimated as part of a Dinosauria tree (as done in our 2016 paper and in our reanalyses using the N=961 trees), such a comparison can be valid, but only because this specific tree comprises a taxonomic sample (N=87) that spans a similar range to that in our larger trees. Unsurprisingly, this is the only tree analysed by Bonsor et al. that supports the downturn model (please see screenshot from Bonsor et al. (2020) here):

The other trees analysed are not the equivalent to the three main subclades (Ornithischia [minus hadrosauriforms and ceratopsids], Sauropodomorpha, and Theropoda) modelled in our 5-group model (here and in our 2016 paper): Arbour dataset, Ankylosauria (N=57); Chiba dataset, Ceratopsia (N=30); CruzadoC dataset, Hadrosauriformes (N=62); Mallon dataset, Chasmosaurinae (N=27); Raven dataset, Stegosauria (N=23); Thompson dataset, Ankylosauria (N=50); GonzalezR, Titanosauriformes (N=76); and Cau, Coelurosauria (N=141).

Despite constituting the bulk of their trees, none of the ornithischian subclades analysed by Bonsor et al. can be considered equivalent to the subclade modelled in our analyses (all ornithischians excluding hadrosauriforms and ceratopsids), particularly in terms of the taxonomic coverage. Such sparsely or narrowly sampled subclades cannot be treated as the equivalent of a more widely and completely (given the phylogenetic taxonomic sampling available from the literature) sampled tree. Rather surprisingly, three of the datasets analysed by Bonsor et al. (Chiba, CruzadoC, Mallon) are focused on the two subclades (Hadrosauriformes and Ceratopsidae) for which we did not find significant quadratic time effects. Thus, it is misleading to suggest that a lack of strong support for the quadratic model in these subclades – i.e. hadrosauriforms and ceratopsians – means that our selection of the quadratic model for Ornithischia (a larger clade with a longer evolutionary history) is questionable.

Similarly, the dataset representing Theropoda in the analyses of Bonsor et al. (Cau dataset) only covers Coelurosauria, which while making up a large portion of Theropoda, is restricted to the latter half of the evolutionary history of theropods, starting with sometime in the mid- to late Jurassic Period. One would not expect to detect the same slowdown or downturn effects within the time period covered by this tree, compared to our tree covering the entirety of the theropod clade spanning their total evolutionary history.

And if that is not the case, then it is the author's job to convince readers.

No, it was the job of Bonsor et al. to reanalyse our study fairly and not misrepresent published work, rather than to make misleading comparisons using sub-clades to represent whole clades (e.g., Ankylosauria as the equivalent of Ornithischia).

This will require a much more reflective and circumspect approach than is currently manifested in the paper.

We believe the need to be reflective should be directed at Bonsor et al. more than at us. Our 2016 paper takes heterogeneity in diversification dynamics into account. We also tested numerous scenarios to test whether the quadratic effect could be explained by other factors including effects of uneven sampling and biases in the model itself – much of this is in the SI of the 2016 paper. The wording of the 2016 paper is also much more careful than our critics suggest (as we've demonstrated above with direct quotes and screenshots of our 2016 paper). As we've pointed out already, there is a lot of perceived disagreements based on paraphrasing of what we wrote and an obsession over wording. Below is a good example of this from Bonsor et al.:

In general, our results agree with those of Sakamoto *et al.* [16] but we disagree in our interpretation of those results. As described above, Sakamoto *et al.* [16] selected their best models based on a DIC number

The main issue that Bonsor et al. have with our 2016 paper is not the statistical results, but in its interpretation, or rather, *how we worded our confidence in our interpretations.*

downturn model. While this is a valid methodological choice, and differences in opinion about model selection procedures are common, we argue that as their selection of the downturn model as the 'best' model was methodologically equivocal it is unfair to say there is 'overwhelming support' for a downturn in dinosaur speciation rates prior to the Late Cretaceous [16, p. 5036]. Our conclusions also

We believe differences in interpretations are unavoidable and there are valid debates to be had. There are also valid points of concern that have been raised following the publication of our paper, that we have addressed and tested repeatedly (post publication peer-review in work!) – but our results stand even after such additional tests.

Additionally, we understand the criticism of ascribing too much confidence to something that may be uncertain, and we agree to some extent. Having said that, the problem with Bonsor et al.'s paper is that their criticism of how we worded our confidence in our interpretations of the results, is largely based on a series of gross misrepresentations of our work. We have outlined this elsewhere in this response and in the revised manuscript.

Finally, Bonsor et al. question the utility of an entire field of study (phylogenetic comparative methods) without providing adequate assessment of the methods themselves. This is a serious criticism that should have warranted more careful assessments of phylogenetic methods.

>"which, for some reason, is the model formulation that Bonsor et al. [1] chose to test instead"

>"...as Bonsor et al rightly point out (following our explicit and repeated statements...)"

>>>This language comes across as being pejorative and doesn't seem appropriate to me if the job is to convince using reasoned and compelling arguments.

We disagree. The fact of the matter is that Bonsor et al. misrepresented our work and criticised our 2016 paper for failing to address certain issues, while completely missing the fact that we did in fact address these issues. Our comment here is not pejorative but simply stating the fact that Bonsor et al. misrepresented our work.

>"When the group-wise model formulation is used... then the differences in DIC are indisputably in favour of the Time2 model, thus providing overwhelming support for this model of diversification through time"

>>>I see what the authors are trying to say here. However, the group-wise model also posits that some of the youngest groups (ceratopsian, hadrosauroids) show a different pattern that goes in the opposite direction.

As stated above and in the original 2016 paper, we have discussed this heterogeneity in detail as well as in the abstract (please see screen shot above). However, hadrosauriforms and ceratopsians are only two relatively small subclades within the larger trees used in our 2016 study. In fact, separating out ceratopsians may not even have been necessary given that their parameters can be non-significant depending on the data tested. In any case, teasing their effects out allowed us to determine a strongly significant quadratic effect in our model for the vast majority of dinosaurs. Again, this heterogeneity has been repeatedly communicated throughout the 2016 paper. *We did not hide this at all.*

This is relevant to interpretation of this statistical model result. Namely, that the model might then -not- be interpreted as providing 'overwhelming' support for a diversification decline across all Dinosauria.

This argument was made by Bonsor et al, and I feel that the authors here have failed to get to grips with it in an appropriately circumspect manner. See e.g. their title and various other overly-strong statements of unilateral interpretation.

The ambiguity of the support for a quadratic model over a square-root model found by Bonsor et al. is based on a simple Dinosauria-wide model in which group-wise differences in diversification dynamics were not considered. This is something they have incorrectly criticised us of doing even though our interpretations were based on a group-wise model and we have repeatedly discussed heterogenous diversification dynamics, starting with the abstract to our 2016 paper.

Once group-wise differences are considered, there is still 'overwhelming support' for a quadratic model over a linear temporal trend model or the square root time model (which is favoured through DIC comparisons, not parameter significance).

>"The additional trees used in their analyses are very small (N = 23-141), with fewer species sampled per clade, compared to those we used (N = 420 and 614).

>>>The text starts 'species per clade', but the N values given for the original study of Sakamoto et al are not 'per clade values'. They are 'all Dinosauria' values. So, of course the numbers are large compared to the numbers used in the analyses of dinosaurs subclasses presented by Bonsor et al. However clearly this is misleading as currently phrased. See my comment above about explanations of samples sizes and effective samples sizes.

Again, model comparisons in the original 2016 study were conducted on DIC values *based on the entire tree*. The group-wise model parameters were conditioned *on the basis* of these tree-wide model fitting procedures. Likewise, the DIC scores of Bonsor et al. are based on their smaller trees. Since the models are selected based on DIC scores, the sample size that matters are the sample sizes for the DIC estimation, which is the total sample sizes of the modelled datasets. In this sense, all of Bonsor et al.'s DIC scores are based on smaller datasets and pointing this out is not misleading but stating a fact.

I feel it is likely, even after a 'fair' comparison, that the analyses of Sakamoto et al have larger effective sample sizes than the analyses of Bonsor et al.

We do not understand this comment; it seems to be a simple statement that our trees were larger, as outlined elsewhere.

Nevertheless, Sakamoto et al need to demonstrate this in a more transparent way.

We believe our original 2016 paper was clear and transparent in respect to the reviewers' comments (see the multitude of screenshots above). Likewise, we believe our manuscript here is clear and transparent.

>"In fact there is an even large meta tree..."

>>>That's true, it is large. Have the authors looked at the timescale phylogeny and which taxa are controlling the divergence times though? There are various species in there that have essentially unknown phylogenetic affinities (based on comparative anatomical and primary phylogenetic literature). Those taxa sometimes are placed in phylogenetic positions that are deeply nested compared to what you'd guess based on their times of appearance.

This has a big effect on the divergence times found by Lloyd et al. Sure, OK, it's a tree that exists and could be analysed. But the fact that many of the divergence times are pushed to surprisingly old ages is probably artifactual and I believe it would tend to bias in analysis in favour of the hypothesis of Sakamoto et al.

If the reviewer could kindly identify these rogue taxa to us, then we would happily rerun these analyses with these problematic taxa removed. We believe it is only fair that criticisms are precise so that we can be directed appropriately to how we should address this. However, our models do not depend on divergence times but on the relationships between node count and time. That is, the exact dates of the divergences do not really matter in our models. The expectation is that this will have minimal effects. Removing such taxa might even favour the down-turn model even more rather than diminish it.

Having said that, the objective of this current manuscript is not to test whether the larger tree is suffering from potential bias but to offer a response to the criticisms laid out by Bonsor et al. We believe such an assessment would be beyond the scope of the current study. In any case, this sort of request is a form of “moving goal post” where one constantly requests additional evidence – i.e., “what about this” or “what about that”?

Moreover, results from our new analyses of the N=961 trees are the same as the other trees we analysed in our 2016 paper and these small differences in topology are largely irrelevant.

>"Larger sample sizes offer more statistical power in estimating parameters in complex models of diversification. Conversely, smaller sample sizes lack the appropriate statistical power to differentiate between competing models. This is especially so for trees with $N < 50$. Simulations have demonstrated that simple evolutionary parameters like heritability"

>>>Sure, I agree with the principle that large sample sizes give more statistical power. However, the text is very generic. It asserts that variation in sample size within the range of the analyses of Bonsor et al will be important in the specific example. But doesn't provide evidence from simulations that are relevant to the current estimation task.

We do not believe simulations are necessary. The responsibility was for Bonsor et al. to demonstrate that sub-clades have enough statistical power to justify direct comparisons with the larger trees of our 2016 analyses. More crucially, they should have demonstrated that sub-clades with narrower taxonomic scope will recover the same patterns of diversification as observed in a more inclusive clade spanning a longer evolutionary history. Or indeed, they should have been transparent about the fundamental differences in the taxonomic samples between their trees and ours – this information is only apparent once one digs into the supplementary data.

Instead, it refers to estimating heritability, which is really quite a different inference problem.

It is the same inference problem as far as model parameter fit is concerned.

It also lacks nuance. For example, greater statistical power is particularly important when effect sizes are small. So if there is 'overwhelming' support (i.e. large effect sizes), then we may not expect this to be as much of a problem. Note that I'm not saying that the authors are wrong. Instead, I'm saying that the number 'N=50' seems to have been chosen arbitrarily without

much rationale behind it. I don't think this way of making the argument sets a good example for others to follow.

The N=50 size is based on a series of simulations cited in our manuscript, but this point is largely irrelevant now.

>>>Also, what is the effective sample size for this type of analysis. Cecile Ané did some nice work showing the effective sample sizes for phylogenetic comparative methods could be much lower than the number of species samples, and that the effective sample size increases initially rapidly when few species have been sampled, and then slows down, with new additions of species adding relatively little new information. Of course, she was looking at a different estimation problem. I only write this here to indicate that it isn't possible to say a priori what sample size is needed without further evaluation of method performance. That evaluation has not been done, to my knowledge, but could provide some of the best support for the authors' preferred hypothesis.

We no longer focus on sample sizes per se. However, a collection of inconclusive model selections on smaller sub-clades does not mean a larger more complete dataset should also be inconclusive. This counters the common argument that incomplete sampling is an issue.

>"In fact, Bonsor and colleagues [1] could not differentiate between the null model and the two competing models in the majority of the trees they analysed, indicating that their sample sizes do not offer sufficient statistical power".

>>>There's no way to approach this other than to say that this reasoning is clearly flawed from the most basic level of statistics. The fact that the null hypothesis cannot be rejected could -either- reflect low statistical power, as argued here, -or- that there is no evidence for the alternative hypothesis. Please remove or re-phrase this specific sentence.

We no longer discuss sample size.

>"Additionally, counting the number of nodes on smaller trees inevitably leads to a narrower range of speciation events (Fig. 1)"

>>>The fact that smaller trees have fewer nodes is self-evident. Please replace 'narrower range' with 'smaller count' or similar.

We do not understand this comment. Yes, the trees have fewer nodes distributed over a narrower range. This means that there isn't much variation to model. It is less likely to be able to differentiate between the three models when node count is distributed over a narrower range and varying little. We believe 'narrower range' is appropriate to describe the distribution of the Y variable in a regression context.

>"...at least three of their trees suffer from a lack of variation on the main predictor variable, Time"

>>>Agreed.

>"Thus, arguing that one model is not discernible from another, or indeed even from conclusive. In fact, such results suffer from severe sampling biases".

>>>'Bias' refers to systematic effects that would incorrectly prefer one model over others. That isn't the case here. The arguments seem to refer to 'error' or 'uncertainty' — factors that harm the ability to differentiate between models (or 'noise'). So it may be better to say 'severe sample deficiencies'.

We have amended the relevant wording in our revised manuscript.

>>>The next section of the paper repeats the analyses of Sakamoto et al. This time they use a recent distribution of supertrees that includes approx 1000 species. They find that the group-wise model is strongly favoured, and includes a general tendency for rates to decrease through time, barring contrarian increases in some of the youngest subclades.

>>>See above, and comments in Bonsor et al. This model doesn't imply that dinosaurs 'lost their ability to speciate' (paraphrasing the conclusions of Sakamoto et al). But that is the conclusion for the current work.

Please refrain from paraphrasing our written work, taken out of context. The actual wording we used in our 2016 paper was more nuanced and careful than this paraphrased statement. Additionally, we do not mention in our current manuscript, that our model implies that dinosaurs 'lost their ability to speciate' we mention that our model supports a 'diversification decline' once in the text and once in the title. This is a factual statement. Our model does indeed support a 'diversification decline'. But this does not mean that 'dinosaurs lost their ability to speciate'. Please point us to the exact location where we implied this.

>"The results presented by Bonsor et al. [1], and claimed to refute our earlier analysis [2], do not offer any conclusive evidence. Their analyses suffer from: 1) an over-simplified model that is not comparable to our group-wise model; and 2) lack of statistical power owing to the extremely small sample sizes. We also demonstrate using 100 large trees (N = 961) of dinosaurs that the diversification decline model [2] is overwhelmingly supported when newer and better-sampled trees are used."

>>>I will be convinced by this once: (1) The authors provide clearer explanations and comparisons of effective sample size.

We no longer focus on sample size.

(2) Ideally include a more well-reasoned explanation of their N = 50 sample size threshold. We already explained this and cited our sources in our manuscript. But it is largely irrelevant now.

And (3) Acknowledge the other points made by Bonsor et al regarding (i) Model interpretation (that some groups show a contrarian pattern)

This point was repeatedly made in the original 2016 paper. We have revised our manuscript to point out the fact that Bonsor et al. are incorrect in their criticism as we have addressed this in extensive detail in our 2016 paper and even qualify our statements in the abstract.

and (ii) Theoretically non-sensible intercept terms.

Again, we described and discussed this in our 2016 paper. However, we have revised our manuscript to reiterate this point but also to clarify that Bonsor et al. have erroneously criticised our work on this point.

Reviewer: 2

Comments to the Author(s)

The manuscript by Sakamoto et al. provides a series of critics to the paper recently published by Bensor et al. (2020) in this same journal. I consider that this is a healthy exercise of implementation of the scientific method. The reasons stated by Sakamoto et al. as critics to the Bensor et al.'s analyses are soundly and overlying any statistically-based methodology. I would like to see a slightly more detailed discussion about the simulations conducted by previous authors about the statistical power of these analysis. Beyond that, I think that Sakamoto et al.'s manuscript is worth of publishing after a few very minor changes indicated in the PDF version of the manuscript.

We no longer discuss sample size but we focus on the false equivalency between small subclade trees and our whole-clade tree.

===PREPARING YOUR MANUSCRIPT===

If you have been asked to revise the written English in your submission as a condition of publication, you must do so, and you are expected to provide evidence that you have received language editing support. The journal would prefer that you use a professional language editing service and provide a certificate of editing, but a signed letter from a colleague who is a native speaker of English is acceptable. Note the journal has arranged a

number of discounts for authors using professional language editing services (<https://royalsociety.org/journals/authors/benefits/language-editing/>).

===PREPARING YOUR REVISION IN SCHOLARONE===

Journal Name: Royal Society Open Science

Journal Code: RSOS

Online ISSN: 2054-5703

Journal Admin Email: openscience@royalsociety.org

Journal Editor: Andrew Dunn

Journal Editor Email: openscience@royalsociety.org

MS Reference Number: RSOS-202143

Article Status: SUBMITTED

MS Dryad ID: RSOS-202143

MS Title: Overwhelming support for diversification decline in dinosaurs

MS Authors: Sakamoto, Manabu; Benton, Michael; Venditti, Chris

Contact Author: Manabu Sakamoto

Contact Author Email: msakamoto@lincoln.ac.uk

Contact Author Address 1: Philip Lyle Building

Contact Author Address 2:

Contact Author Address 3:

Contact Author City: Reading

Contact Author State:

Contact Author Country: United Kingdom of Great Britain and Northern Ireland

Contact Author ZIP/Postal Code: RG6 6BX

Keywords: dinosaurs, phylogenetic comparative methods, GLMM, diversification rate, diversification decline

Abstract:

EndDryadContent